# Improving expressivity in Link Prediction with GNNs via the Shortest Path

## Abstract

Graph Neural Networks (GNNs) often fail to capture the link-specific structural patterns essential for accurate link prediction, since their node-centric message passing might overlook the subgraph structures connecting two nodes. Prior attempts to inject such structural context either suffer from high computational cost or rely on oversimplified heuristics (e.g., common neighbor counts) that cannot capture multi-hop dependencies. We propose SP4LP (Shortest Path for Link Prediction), a new framework that integrates GNN-based node encodings with sequence modeling over shortest paths. Specifically, SP4LP first computes node representations with a GNN, then extracts the shortest path between each candidate node pair and processes the sequence of node embeddings with a sequence model. This design allows SP4LP to efficiently capture expressive multi-hop relational patterns. Theoretically, we show that SP4LP is strictly more expressive than both standard message-passing GNNs and several leading structural feature methods, positioning it as a general and principled framework for link prediction in graphs. Empirically, SP4LP sets a new state of the art on many standard link prediction benchmarks.

## 1 Introduction

Graph Neural Networks (GNNs) are widely adopted for link-level tasks such as link prediction (Zhang & Chen, 2018; Lü & Zhou, 2011; Zhou, 2021), link classification (Rossi et al., 2021; Wang et al., 2021; Cheng et al., 2025) and link regression (Liang et al., 2025; Dong et al., 2019) with applications spanning recommender systems (Ying et al., 2018), knowledge graph completion (Nickel et al., 2015), and biological interaction prediction (Jha et al., 2022).

Despite their popularity, standard GNNs struggle to accurately represent links, as they typically construct link embeddings by aggregating the representations of the two endpoint nodes. This node-centric strategy leads to a key limitation: structurally distinct links may be mapped to the same representation when their endpoints are automorphic (Srinivasan & Ribeiro, 2019; Chamberlain et al., 2023; Zhang et al., 2021). For example, in the graph of Figure 1, links $(v, u)$ and $(v, u')$ yield identical representations under any standard GNN, even if one pair shares a common neighbor and the other does not. This issue, known as the automorphic node problem (Chamberlain et al., 2023), highlights a fundamental expressivity bottleneck in message-passing schemes for link representation.

To address this, several methods enhance GNNs with structural features (SFs), which can be broadly classified into three paradigms (Wang et al., 2024): SF-then-GNN, which injects structural context into the graph before message passing (e.g., SEAL (Zhang et al., 2021), NBFNet (Zhu et al., 2021)); SF-and-GNN, which computes SFs and node embeddings in parallel (e.g., Neo-GNN (Yun et al., 2021), BUDDY (Chamberlain et al., 2023)); and GNN-then-SF, which applies message passing once to compute node representations and then combines them using task-specific structural context (e.g., NCN and NCNC (Wang et al., 2024)).

While SF-then-GNN methods are expressive, they are computationally inefficient, often requiring subgraph extraction or retraining per link. SF-and-GNN models are efficient but rely on predefined heuristics, limiting their ability to capture rich relational patterns. GNN-then-SF approaches offer a compelling trade-off between expressivity and scalability, but current methods in this class, i.e., NCN and NCNC, still rely on overlap between the neighborhoods of the endpoints: NCN uses

common neighbors, while NCNC also leverages predicted second-order common neighbors. In the absence of such overlapping neighbors, they revert to standard GNN behavior and lose expressive power.

In this paper we propose SP4LP, a novel method in the GNN-then-SF paradigm that combines high expressiveness with computational efficiency. SP4LP constructs a path-aware representation by incorporating the embeddings of all nodes along the shortest path connecting the two endpoints. These node embeddings, obtained via a base GNN, are then processed as a sequence using a dedicated sequence model, such as a Transformer (Vaswani et al., 2017), LSTM (Hochreiter & Schmidhuber, 1997), or an injective summation function (Xu et al., 2019).

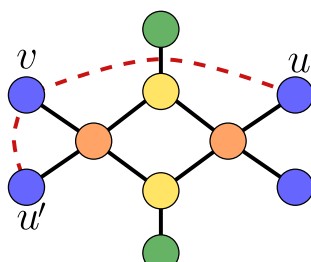

The first key advantage of SP4LP is its **expressiveness**. Unlike structural features such as common neighbors, which may not exist for many node pairs and can lead to degenerate cases in sparse graphs, the shortest path is always defined between any two nodes if the graph is connected. Shortest paths are more broadly defined, as common neighbors imply a path, but not vice versa. Moreover, since the embeddings of the nodes along the path are generated through message passing, they implicitly encode the broader local structure surrounding the link.

Figure 1: Links $(v, u)$ and $(v, u')$ have different structural roles within the graph, yet a GNN assigns them identical representations.

This richer structural context enables SP4LP to distinguish non-automorphic links even when their endpoints are automorphic, thereby overcoming the automorphic node problem. We formally prove that SP4LP is strictly more expressive than some existing approaches.

SP4LP is **efficient** and **scalable**. The message-passing step is performed only once on the entire graph, and the shortest path computation is a preprocessing step. Unlike SF-then-GNN methods such as SEAL or NBFNet, SP4LP avoids costly per-link subgraph extraction or online traversal during inference, allowing it to scale to large graphs and high-throughput settings.

Moreover, SP4LP is a **general and flexible framework**. It can be instantiated with any GNN architecture to compute node embeddings (e.g., GCN (Kipf & Welling, 2016a), GAT (Velickovic et al., 2017), GraphSAGE (Hamilton et al., 2017b)), and supports a range of sequence models for encoding the path structure, from lightweight aggregators to fully expressive recurrent or attention-based models.

Finally, SP4LP achieves **state-of-the-art performance** across several benchmark datasets. Under the challenging HeaRT evaluation protocol (Li et al., 2023), it consistently outperforms existing link prediction methods while maintaining competitive inference speed and low memory usage.

These properties make SP4LP a principled and practical solution for learning expressive link representations in real-world graph learning scenarios.

## 2 PRELIMINARIES

**Definition 2.1** (*graph*). *A **graph** is a tuple $G = (V, E, \mathbf{X}^0)$ where $V = \{1, \ldots, n\}$ is a set of nodes, $E \subseteq V \times V$ is a set of edges and $\mathbf{X}^0 \in \mathbb{R}^{n \times f}$ is the node features matrix. To each graph is associated an adjacency matrix $\mathbf{A} \in \{0, 1\}^{n \times n}$ with $\mathbf{A}_{i,j} = 1$ if and only if $(i, j) \in E$. In this work, we consider simple, finite and undirected graphs.*

**Definition 2.2** (*message passing*). *Let $G = (V, E, \mathbf{X}^0)$ be a graph. In **message passing** scheme, representation of nodes $v \in V$ is iteratively updated as follows:*

$$\mathbf{x}_v^0 = \mathbf{X}_{[v,:]}^0 \tag{1}$$

$$\mathbf{x}_v^l = \text{UPDATE}\left(\mathbf{x}_v^{l-1}, \text{AGGREGATE}\left(\{\mathbf{x}_u^{l-1} \mid u \in N(v)\}\right)\right) \tag{2}$$

*where $N(v)$ is the first-order neighborhood of node $v$.*

Graph Neural Networks (GNNs) are a class of neural architectures that operate on graphs by iteratively updating node representations through the message passing scheme. It has been proven

that GNNs are at most as effective as the Weisfeiler–Lehman (WL) test in distinguishing between graphs (Morris et al., 2019; Xu et al., 2019).

**Definition 2.3** (GNN link representation model). *A **GNN link representation model** $M$ is a class of functions*

$$F : ((u,v), G) \mapsto \mathbf{x}_{(u,v)} \in \mathbb{R}^d \tag{3}$$

*which maps node pairs in $(u,v) \in V \times V$ to vector representations using the message passing scheme defined in Definition 2.2.*

Note that the pair $(u,v)$ belongs to $V \times V$, meaning that we compute a representation for any node pair, what we refer to as a *link*, regardless of whether an edge between them exists in $E$. This general definition reflects the nature of the downstream tasks we aim to address once the link representation is available, most notably, link prediction, where the objective is to estimate the likelihood of a connection between arbitrary node pairs. To this end, the model learns representations for all possible pairs, not just those connected by an edge. A widely adopted approach for learning such representations is what we refer to as a *pure GNN*, defined as follows:

**Definition 2.4** (*pure GNN*). *A **pure GNN** model calculates representation $\mathbf{x}_{(u,v)} \in \mathbb{R}^d$ for each pair of nodes $(u,v)$ with $u, v \in V$ as follows:*

$$\mathbf{x}_{(u,v)} = g(\mathbf{x}_u^L, \mathbf{x}_v^L) \tag{4}$$

*where $g$ is an aggregation function and $\mathbf{x}_u^L, \mathbf{x}_v^L$ are the node representation of $u$ and $v$ learned by $L$ layers of message passing as defined in Definition 2.2.*

Pure GNNs are inherently limited in terms of expressiveness. In particular even when the base GNN is the most powerful, they may assign the same representation to structurally different links. Consider, for example, the graph in Figure 1: the colors of the nodes indicate the colors produced by the WL algorithm; thus, using a most powerful GNN nodes $u, u'$ will be assigned to the same representation. As a result, no matter how expressive the aggregation function $g$ is, the representations of the pairs $(v,u)$ and $(v,u')$ will be identical. However, the links $(v,u)$ and $(v,u')$ have different roles within the graph structure. We provide a formal definition of what it means for two links to be different.

**Definition 2.5** (*node permutation*). *A **node permutation** $\pi : \{1, \ldots, n\} \to \{1, \ldots, n\}$ is a bijective function that assigns a new index to each node of the graph. All the $n!$ possible node permutations constitute the permutation group $\Pi_n$. Given a subset of nodes $S \subseteq V$, we define the permutation $\pi$ on $S$ as $\pi(S) := \{\pi(i) | i \in S\}$. Additionally, we define $\pi(\mathbf{A})$ as the matrix $\mathbf{A}$ with rows and columns permutated based on $\pi$, i.e., $\pi(\mathbf{A})_{\pi(i),\pi(j)} = \mathbf{A}_{i,j}$.*

**Definition 2.6** (*automorphism*). *An **automorphism** on the graph $G = (V, E, \mathbf{X}^0)$ is a permutation $\sigma \in \Pi_n$ such that $\sigma(\mathbf{A}) = \mathbf{A}$. All the possible automorphisms on a graph constitute the automorphism group $\Sigma_n^G$.*

**Definition 2.7** (*automorphic nodes*). *Let $G = (V, E, \mathbf{X}^0)$ be a graph and $\Sigma_n^G$ its automorphism group. Two nodes $u, v \in V$ are said to be **automorphic nodes** ( $u \simeq v$ ) if:*

$$\exists \sigma \in \Sigma_n^G \quad s.t. \quad \sigma(\{u\}) = \{v\}. \tag{5}$$

**Definition 2.8** (*automorphic links*). *Let $G = (V, E, \mathbf{X}^0)$ be a graph and $\Sigma_n^G$ its automorphism group. Two pairs of nodes $(u,v), (u',v') \in V \times V$ are said to be **automorphic links** ( $(u,v) \simeq (u',v')$ ) if:*

$$\exists \sigma \in \Sigma_n^G \quad s.t. \quad \sigma(\{u,v\}) = \{u',v'\}. \tag{6}$$

**Proposition 2.9.** *Pure GNN methods suffer from the automorphic node problem, i.e., for any graph $G = (V, E, \mathbf{X}^0)$, for pairs of links $(u,v), (u',v') \in V \times V$ such that there exist $\sigma_1 \in \Sigma_n^G$ and $\sigma_2 \in \Sigma_n^G$ with $\sigma_1(u) = u'$ and $\sigma_2(v) = v'$, $\mathbf{x}_{(u,v)} = \mathbf{x}_{(u',v')}$, independently whether $(u,v)$ and $(u',v')$ are isomorphic, i.e, whether exist $\sigma \in \Sigma_n^G$ with $\sigma(\{u,v\}) = \{u',v'\}$.*

This limitation is well-known in the literature (Chamberlain et al., 2023; Zhang et al., 2021). Importantly, it does not arise from the expressiveness bounds of GNNs, which are constrained by the WL test. Even considering higher-order GNNs, i.e., $k$-GNN (Morris et al., 2019), automorphic nodes will be assigned to the same representation as the $k$-WL algorithm preserves graph automorphisms

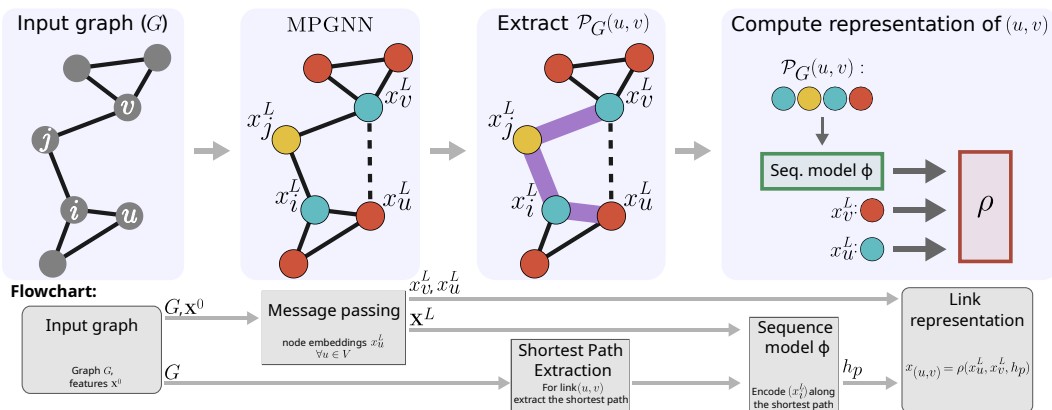

Figure 2: Overview of the SP4LP framework. First, a GNN is used to compute contextualized embeddings for all nodes in the graph. Then, for each target link, the shortest path connecting the two endpoints is extracted. The embeddings of the nodes along this path are passed to a sequence model (e.g., Transformer or LSTM) to compute a path-aware link representation. Below, we also include a flowchart summarizing the full SP4LP pipeline, highlighting each computational step and its inputs/outputs.

for every $k$ (Lichter et al., 2025; Dawar & Vagnozzi, 2020; Cai et al., 1992). Thus, no standard GNN can distinguish non automorphic links composed by automorphic nodes.

To tackle this, several models have been proposed that enhance message passing by incorporating structural features (Wang et al., 2024; Zhu et al., 2021; Chamberlain et al., 2023; Wang et al., 2022; Zhang et al., 2021), thereby increasing the expressive power of the resulting link representations. We provide a formal definition of what it means for one link representation model to be more expressive and strictly more expressive than another.

**Definition 2.10** (*more expressive*)**.** *Let $M_1$ and $M_2$ be two link representation models (Def. 2.3). $M_2$ is **more expressive** than $M_1$ ($M_1 \preceq M_2$) if, for any graph $G = (V, E, \mathbf{X}^0)$ and any pair $(u, v), (u', v') \in V \times V$ with $(u, v) \not\simeq (u', v')$:*

$$\exists F_1 \in M_1 : F_1((u, v), G) \neq F_1((u', v'), G) \Rightarrow \exists F_2 \in M_2 : F_2((u, v), G) \neq F_2((u', v'), G). \quad (7)$$

**Definition 2.11** (*strictly more expressive*)**.** *Let $M_1$ and $M_2$ be two link representation models (Def. 2.3). We say that $M_2$ is **strictly more expressive** than $M_1$ ($M_1 \prec M_2$) if:*

- *$M_2$ is more expressive than $M_1$ (Def. 2.10), and*

- *there exists a graph $G = (V, E, \mathbf{X}^0)$ and a pair of links $(u, v), (u', v') \in V \times V$ with $(u, v) \not\simeq (u', v')$ such that:*

$$\forall F_1 \in M_1 : \quad F_1((u, v), G) = F_1((u', v'), G)$$
$$and \quad \exists F_2 \in M_2 : \quad F_2((u, v), G) \neq F_2((u', v'), G).$$

In the following section, we introduce our model SP4LP and demonstrate its improved expressive power in distinguishing structurally different links.

## 3 RELATED WORK

GNNs have been extensively used for link representation tasks. In standard approaches like Graph Autoencoders (GAE) (Kipf & Welling, 2016b), node embeddings are computed via message passing, and a simple decoder (e.g., inner product followed by a sigmoid) predicts link existence. While efficient, these models exhibit limited expressiveness (Zhang et al., 2021; Chamberlain et al., 2023), primarily due to their inability to capture rich structural patterns beyond immediate neighborhoods. This limitation has motivated the integration of explicit structural information into GNN-based models.

**Incorporating Structural Information into GNNs.** To overcome expressiveness bottlenecks, several methods augment GNNs with structural features. Neo-GNN (Yun et al., 2021) injects hand-crafted features into the message-passing process, while ELPH (Chamberlain et al., 2023) and its scalable variant BUDDY employ MinHash and HyperLogLog sketches to capture multi-hop patterns, with BUDDY precomputing sketches offline. NCN (Wang et al., 2024) aggregates embeddings from endpoint nodes and their common neighbors, and NCNC extends this by predicting missing neighbors before reapplying NCN. NBFNet (Zhu et al., 2021) instead aggregates information over all paths between node pairs via Bellman-Ford-inspired recursive functions. Differently from these approaches, our method focuses explicitly on the shortest path between node pairs, using a sequence model to capture dependencies along this path. This results in more focused and interpretable representations while avoiding the inefficiencies of modeling broader multi-hop neighborhoods or exhaustive path sets. Beyond GNN-based approaches, PROXI (Tola et al., 2025) proposes a hybrid proximity-based framework built on handcrafted structural indices combined with a non-GNN predictor (XGBoost). While conceptually related through the use of structural signals, PROXI fundamentally differs from GNN-based link representation models.

**Enhancing GNN Expressiveness through Positional and Structural Encoding.** Positional encodings further enrich GNN expressiveness. PEG (Wang et al., 2022) integrates Laplacian-based encodings into message passing, weighting neighbors by positional distances. SEAL (Zhang et al., 2021) extracts $h$-hop enclosing subgraphs and labels nodes via the DRNL scheme before applying a GNN. While expressive, these methods struggle to scale due to subgraph extraction overhead. In contrast, our model achieves expressiveness by operating directly on compact, informative shortest-path sequences, enabling better scalability without sacrificing representational power.

**Shortest-Path Structures in Graph Learning.** Shortest-path information has also proven effective in tasks beyond link prediction, such as graph classification (Ying et al., 2021; Airale et al., 2025) and node classification on heterophilous graphs (Li et al., 2020). These works highlight the power of shortest-path structures across graph learning domains. Building on this insight, our model directly leverages shortest-path sequences for link representation, showing that this structure is particularly effective when combined with modern sequence models.

**Comparison with GDGNN** Geodesic GNN (GDGN) Kong et al. (2022) is the closest method to ours in that it also extracts shortest-path information after a single GNN run. However, our approach differs in three key aspects. (i) Path modeling: GDGNN applies permutation-invariant pooling over the path nodes, losing the order and direction of the geodesic. SP4LP instead treats the shortest path as an ordered sequence and processes it with a sequence model, enabling the capture of structural patterns that pooling cannot represent. (ii) Objective: GDGNN is a broad framework aimed at improving WL-level expressiveness; SP4LP is a simple, task-specific model for structural link representation. (iii) Scalability: GDGNN includes additional geodesic modules and subgraph reasoning, whereas SP4LP keeps inference lightweight by decoupling the global GNN run from per-query path processing.

# 4 SP4LP: AN EXPRESSIVE GNN-THEN-SF MODEL FOR LINK REPRESENTATION

Existing GNN link representation models that leverage Structural Features (SF) fall into three categories (Wang et al., 2024): **SF-and-GNN**, which compute SF and GNN embeddings separately and then combine them (e.g., Neo-GNN (Yun et al., 2021), BUDDY (Chamberlain et al., 2023)); **SF-then-GNN**, which augment the graph with SF before applying GNN (e.g., SEAL (Zhang & Chen, 2018), NBFNet (Zhu et al., 2021)); and **GNN-then-SF**, which compute GNN embeddings first and then aggregate them using SF (e.g., NCN, NCNC (Wang et al., 2024)).

In this work, we adopt the GNN-then-SF paradigm, which combines the scalability advantages of applying message passing only once, as in the SF-and-GNN setting, with the expressiveness typical of SF-then-GNN approaches, due to the ability to aggregate over task-specific sets of node representations. To date, the only existing models that follow this paradigm are NCN and NCNC. For a more in-depth discussion on the differences between our method, SP4LP and NCN, refer to Section 4.1. In the following, we introduce the necessary definitions, formally describe the model, and present theoretical results characterizing its expressive power.

**Definition 4.1** (*path*). *Let $G = (V, E, \mathbf{X}^0)$ be a graph and $u, v \in V$ to nodes. A **path** in $G$ from $u$ to $v$ is a sequence of nodes $P = (u_0, u_1, \ldots, u_k)$ with (i) $u_i \in V$ for all $i = 0, \ldots, k-1$, (ii) $u_0 = u$ and $u_k = v$, (iii) $(u_i, u_{i+1}) \in E$ for all $i = 0, \ldots, k-1$, and (iv) all nodes in the sequence are distinct (i.e., $u_i \neq u_j$ for all $i \neq j$). The length of a path $P$, $len(P)$ is the number of edges it contains.*

**Definition 4.2** (*shortest path length*). *Let $\mathcal{P}_G(u, v)$ denote the set of all paths from $u$ to $v$ in $G$. The **shortest path length** $d_G(u, v)$ is the minimum length among all paths i.e., $d_G(u, v) = \min_{P \in \mathcal{P}(u,v)} len(P)$.*

**Definition 4.3** (*shortest path*). *A **shortest path** between $u$ to $v$ in $G$ is any path $P^* \in \mathcal{P}_G(u, v)$ such that $len(P^*) = d_G(u, v)$. The set of all the shortest path from $u$ to $v$ in $G$ is denoted as $\mathcal{P}_G^*(u, v)$.*

Let $G = (V, E, \mathbf{X}^0)$ be a graph, $u, v \in V$. SP4LP is a GNN link representation model (see Definition 2.3) that computes link representation as follows:

$$\text{SP4LP}((u, v), G) = \rho \left( \text{GNN}(u, G), \text{GNN}(v, G), \text{AGG}\left( \left\{ \phi \left( \text{GNN}(u_i, G) \right)_{i=1}^k \mid (u_i)_{i=1}^k \in \mathcal{P}_G^*\{u, v\} \right\} \right) \right) \tag{8}$$

where $k = d_G(u, v)$, $\text{GNN}(u, G) \in \mathbb{R}^d$ is the representation of node $u \in V$ obtained at the final layer of message passing as in Definition 2.2, $\phi : \mathbb{R}^{k \times d} \to \mathbb{R}^d$ is a sequence model on the GNN representations of nodes in the shortest path from $u$ to $v$, AGG is an aggregation function over multiset of shortest paths representations and $\rho : \mathbb{R}^d \times \mathbb{R}^d \times \mathbb{R}^d \to \mathbb{R}^d$ combine the endpoint nodes representations with the shortest paths representation to get a final link representation. Since we consider undirected graphs, we consider $\mathcal{P}_G^*\{u, v\} := \mathcal{P}_G^*(u, v) \cup \mathcal{P}_G^*(v, u)$.

Although shortest paths provide a focused and efficient structural summary, they may lose information carried by alternative routes or larger subgraphs, reducing robustness and emphasizing local rather than global structure. We quantify this trade-off experimentally in Section 5.2, showing that replacing full path embeddings with simple distance information leads to significant performance degradation.

For a graph with $n$ nodes and $m$ edges, single-source BFS (Cormen et al., 2009) runs in $O(n + m)$ time on an adjacency-list representation. Although computing all-pairs shortest paths (APSP) would require $O(n(n+m))$ time, SP4LP does not perform APSP. In our experiments, we compute shortest paths only for the supervised train/validation/test links, which requires at most a small number of BFS traversals. Let $S$ be the number of distinct source nodes among these links; the preprocessing cost is therefore $O(S(n + m))$, typically several orders of magnitude smaller than $O(n(n + m))$. The resulting paths are cached and reused throughout training and inference, so no graph traversal is performed during model training or batched evaluation. In fully-inductive settings where new nodes may appear, one can compute shortest-path trees by running a single BFS per new node, providing distances and parent pointers to all existing nodes at once. This preserves the same per-link inference efficiency and avoids BFS operations per queried link. A more detailed analysis of the overall complexity, including a comparison with prior approaches, is provided in Appendix E. In practice, all shortest paths are precomputed before training, and SP4LP simply retrieves the stored sequences during training and inference. No BFS operations are performed online.

SP4LP is a general and flexible framework: both the underlying GNN used to compute node representations and the sequence model used to process the embeddings along the shortest path can be chosen modularly. For instance, the GNN component can be instantiated with architectures such as GCN (Kipf & Welling, 2016a), GAT (Velickovic et al., 2017) or GraphSAGE (Hamilton et al., 2017b), while the sequence model can range from simple aggregation functions like injective summation as the one proposed in Xu et al. (2019), to more complex architectures such as LSTMs (Hochreiter & Schmidhuber, 1997), GRU (Chung et al., 2014) or Transformers (Vaswani et al., 2017). An overview of SP4LP is illustrated in Figure 2.

The additional structural context given by the sequence of embeddings of nodes within the shortest path enables the model to distinguish links that are otherwise indistinguishable to standard message-passing methods, such as those involving automorphic nodes.

**Proposition 4.4.** SP4LP *does not suffer from the automorphic node problem.*

The proof can be found in Appendix A. As an example of non-automorphic links composed of automorphic nodes that SP4LP can successfully distinguish, consider the links $(v, u)$ and $(v, u')$

shown in Figure 1. While $u' \simeq u'$ via the identity, and $v \simeq u$ via an automorphism induced by a vertical axis of symmetry (i.e., a mirror reflection), the links $(v, u)$ and $(v, u')$ are not automorphic. This asymmetry is captured by the distinct shortest paths between the endpoints: the shortest path from $v$ to $u'$ consists of $v$, and orange node, and $u'$, whereas the shortest path from $v$ to $u$ includes $v$, and orange node, a yellow node, another orange node, and finally $u$. In addition to overcoming this limitation, SP4LP is strictly more expressive than several state-of-the-art message passing methods for link representation learning.

**Theorem 4.5.** SP4LP *is strictly more expressive than Pure GNNs, NCN, BUDDY, NBFnet and Neo-GNN.*

The proof can be found in Appendix A. In the following sections, we complement the theoretical analysis with an extensive experimental evaluation, showing that SP4LP also achieves state-of-the-art performance on standard link prediction benchmarks.

### 4.1 FURTHER COMPARISON WITH NCN

NCN computes the representation of a link by aggregating the GNN embeddings of its endpoints and their common neighbors. However, its reliance on the common neighbor structure makes it particularly vulnerable to graph incompleteness. Moreover, the use of common neighbors as the sole source of structural features results in a critical failure mode: when two nodes share no neighbors, NCN reduces to a pure GNN and can no longer leverage structural information. This situation is far from rare in practice (see Table 1), where a substantial fraction of positive links involve endpoints without common neighbors. In contrast, we propose SP4LP, which incorporates additional pairwise information by encoding the sequence of node embeddings along the shortest path connecting the endpoints. Unlike common neighbors, the shortest path is always defined in connected graphs and captures richer structural patterns, even in sparse or incomplete settings.

Table 1: Fraction of positive test links without common neighbors.

| Dataset | w/o CN |
|---|---|
| Cora | 46% |
| Citeseer | 55% |
| Pubmed | 67% |
| ogbl-collab | 52% |
| ogbl-ddi | 0.04% |
| ogbl-ppa | 10% |
| ogbl-citation2 | 37% |

Moreover, while NCN pools neighbors as unordered sets, SP4LP encodes paths as ordered sequences via LSTMs or Transformers, yielding strictly higher expressiveness (Theorem 4.5).

## 5 EXPERIMENTS

Models that compute link representations can be applied to a wide range of downstream tasks, with link prediction being particularly impactful due to its broad applicability in domains such as recommender systems (Ying et al., 2018), knowledge graph completion (Nickel et al., 2015), and biological interaction prediction (Jha et al., 2022). The link representations produced by GNN methods are used to estimate the probability of existence for each candidate link. To train models for link prediction, existing edges in the graph are treated as positive examples, while negative examples are generated through negative sampling, selecting node pairs that are not connected in the original graph.

In this section, we extensively evaluate the performance of SP4LP on real-world link prediction benchmarks against several baselines. In particular, we use three Planetoid citation networks: Cora, Citeseer, and Pubmed (Yang et al., 2016) as well as two datasets from Open Graph Benchmark Hu et al. (2020), i.e., ogbl-collab and ogbl-ddi. For Cora, Citeseer, and Pubmed, we use a single fixed data split in all experiments. Table 7 in appendix C provides a summary of dataset statistics.

As baseline methods we consider three class of models: 1) **heuristic methods**: Common Neighbors (CN) (Newman, 2001), Adamic-Adar (AA) (Adamic & Adar, 2003), Resource Allocation (RA) (Zhou et al., 2009), Shortest Path (SP) (Liben-Nowell & Kleinberg, 2003), and Katz (Katz, 1953); 2) **Embedding-based methods**: Node2Vec (Grover & Leskovec, 2016), Matrix Factorization (MF) (Menon & Elkan, 2011), and a Multilayer Perceptron (MLP) applied to node features; 3) **Pure GNN methods**: Graph Convolutional Network (GCN) (Kipf & Welling, 2016a), Graph Attention Network (GAT) (Veličković et al., 2018), GraphSAGE (Hamilton et al., 2017a), and Graph Autoencoder (GAE) (Kipf & Welling, 2016b); 4) **Structural Features GNN methods**: SEAL (Zhang

Table 2: MRR and Hits@K (%) results across all datasets, following the HeaRT evaluation setting Li et al. (2023). The top three results for each metric are highlighted using **first**, *second*, and **third**. *OOM* indicates that the model ran out of memory, while *>24h* denotes that the method did not complete within 24 hours. Standard deviations over 5 runs are reported in the appendix B.

| Models | Cora | | Citeseer | | Pubmed | | Ogbl-ddi | | Ogbl-collab | | Ogbl-ppa | | Ogbl-Citation2 | |
|---|---|---|---|---|---|---|---|---|---|---|---|---|---|---|
| | MRR | Hits@10 | MRR | Hits@10 | MRR | Hits@10 | MRR | Hits@20 | MRR | Hits@20 | MRR | Hits@20 | MRR | Hits@20 |
| GCN | 16.61 | 36.26 | 21.09 | 47.23 | 7.13 | 15.22 | 13.46 | 64.76 | 6.09 | 22.48 | 26.94 | 68.38 | 19.98 | 51.72 |
| GAT | 13.84 | 32.89 | 19.58 | 45.30 | 4.95 | 9.99 | 12.92 | 66.83 | 4.18 | 18.30 | *OOM* | *OOM* | *OOM* | *OOM* |
| SAGE | 14.74 | 34.65 | 21.09 | 48.75 | 9.40 | 20.54 | 12.60 | 67.19 | 5.53 | 21.26 | 27.27 | 69.49 | 22.05 | 53.13 |
| GAE | 18.32 | 37.95 | 25.25 | 49.65 | 5.27 | 10.50 | 3.49 | 17.81 | *OOM* | *OOM* | *OOM* | *OOM* | *OOM* | *OOM* |
| SEAL | 10.67 | 24.27 | 13.16 | 27.37 | 5.88 | 12.47 | 9.99 | 49.74 | 6.43 | 21.57 | 29.71 | 76.77 | 20.60 | 48.62 |
| BUDDY | 13.71 | 30.40 | 22.84 | 48.35 | 7.56 | 16.78 | 12.43 | 58.71 | 5.67 | 23.35 | 27.70 | 71.50 | 19.17 | 47.81 |
| Neo-GNN | 13.95 | 31.27 | 17.34 | 41.74 | 7.74 | 17.88 | 10.86 | 51.94 | 5.23 | 21.03 | 21.68 | 64.81 | 16.12 | 43.17 |
| NCN | 14.66 | 35.14 | 28.65 | 53.41 | 5.84 | 13.22 | 12.86 | 65.82 | 5.09 | 20.84 | 35.06 | 81.89 | 23.35 | 53.76 |
| NCNC | 14.98 | 36.70 | 24.10 | 53.72 | 8.58 | 18.81 | *>24h* | *>24h* | 4.73 | 20.49 | 33.52 | 82.24 | 19.61 | 51.69 |
| NBFNet | 13.56 | 31.12 | 14.29 | 31.39 | *>24h* | *>24h* | *>24h* | *>24h* | *OOM* | *OOM* | *OOM* | *OOM* | *OOM* | *OOM* |
| PEG | 15.73 | 36.03 | 21.01 | 45.56 | 4.40 | 8.70 | 12.05 | 50.12 | 4.83 | 18.29 | *OOM* | *OOM* | *OOM* | *OOM* |
| LPFORMER | 16.80 | 34.03 | 26.34 | 51.72 | 9.99 | 21.43 | 13.20 | 62.66 | 7.62 | 21.04 | 40.25 | 84.1 | 24.70 | 57.30 |
| SP4LP (our) | 17.27 | 38.52 | 41.08 | 66.28 | 10.87 | 23.01 | 15.00 | 47.96 | 9.46 | 20.00 | 36.45 | 76.90 | 24.91 | 55.45 |

& Chen, 2018), BUDDY (Chamberlain et al., 2023), Neo-GNN (Yun et al., 2021), NBFNet (Zhu et al., 2021), NCN (Wang et al., 2024), NCNC (Wang et al., 2024) and PEG (Wang et al., 2022).

Importantly, as described in Section 4, SP4LP is a general framework that allows for different choices of both the underlying GNN architecture and the sequence model ($\phi$ Equation 8). In our experimental setting, we treat the choice of GNN and the choice of $\phi$ as hyperparameters, and perform hyperparameter tuning based on validation set performance. Specifically, we explore GCN, GAT, and GraphSAGE as GNN backbones, and LSTM, Transformer, and an injective sum Xu et al. (2019) aggregator as sequence models. Moreover, we choose an MLP for $\rho$ and as AGG we choose to select the first shortest path retrieved by the BFS procedure for computational efficiency. Appendix D provides implementation details, including how shortest paths between node pairs are computed, as well as the hyperparameter configurations used in our experiments. Code to reproduce all experiments is available at[1].

**Evaluation Setting** We evaluate model performance under the more challenging and realistic HeaRT evaluation setting Li et al. (2023). In this setting, each positive target link (i.e., an existing link) is ranked against a carefully selected set of hard negative samples (i.e., non-existing links), providing a more realistic assessment of link prediction performance in practical scenarios. We adopt two standard ranking metrics: Hits@K and Mean Reciprocal Rank (MRR). Following the HeaRT protocol, we report Hits@10 and MRR for Cora, Citeseer, and Pubmed, and Hits@20 along with MRR for ogbl-collab and ogbl-ddi. The same set of negative samples is used across all positive links, as specified in the HeaRT benchmark. The HeaRT evaluation introduces significantly harder negative samples compared to traditional evaluation settings, resulting in a more challenging and realistic benchmark. Li et al. (2023) show that this leads to a substantial performance drop across most models, with GNNs specifically designed for link prediction often being outperformed by simple heuristics or general-purpose GNNs. By adopting this challenging evaluation setting, we ensure a rigorous and meaningful comparison of model performance under conditions that closely resemble real-world applications.

## 5.1 RESULTS ON REAL-WORLD BENCHMARKS

Table 2 presents the performance of SP4LP and the baseline models in terms of MRR and Hits@10 on Cora, Citeseer, and Pubmed, and MRR and Hits@20 on the OGB datasets. SP4LP ranks first in terms of MRR on four out of five datasets and second on the remaining one. The improvements in MRR are often substantial: on Citeseer, for instance, SP4LP achieves a 43% gain over the second-best method, NCN. SP4LP also achieves the best Hits@K score on three out of five datasets. On the Ogbl-Collab dataset, SP4LP is comparable on Hits@20 to the third-best model (SEAL), when accounting for standard deviations (Appendix B). On Ogbl-ddi, where SP4LP performs worse, the

[1]`https://anonymous.4open.science/r/sp4lp-3875/README.md`

lower score can be explained by the lack of node features. Our model benefits from the availability of node features, as it leverages nodes representations obtained via message passing. In settings where such features are absent, like in Ogbl-ddi, the discriminative power of the learned representations is reduced. In addition to achieving the best performance in several datasets, SP4LP is also the most consistent model across all benchmarks. Unlike previous state-of-the-art models such as BUDDY and Neo-GNN, which tend to struggle on datasets like Cora, Citeseer and Pubmed, SP4LP maintains strong performance regardless of dataset characteristics. Overall, these results clearly demonstrate the superiority of SP4LP under the more challenging and realistic HeaRT evaluation setting, confirming its effectiveness for real-world link prediction tasks.

## 5.2 ABLATION STUDY

We perform an ablation study to evaluate the contribution of the main components of our model. In particular, we investigate two simplified variants to understand the importance of node representation learning and sequential modeling along the shortest path between target nodes. **(1) Sequence Model Only**: in this variant, the sequential model operates directly on the raw input features of the nodes along the shortest path, without incorporating node representations learned by the GNN. This setup

Table 3: Ablation study results (%). MRR and Hits@K are reported for two model variants: (1) *Sequence Model Only*, using a sequential model on raw node features, and (2) *GNN + Shortest Path Length*, using GNN representations with path length. The full model SP4LP consistently outperforms both variants. Standard deviations over 5 runs are reported in Appendix B

| Models | Cora | | Citeseer | | Pubmed | |
|---|---|---|---|---|---|---|
| | MRR | Hits@10 | MRR | Hits@10 | MRR | Hits@10 |
| *GNN + SP len.* | 14.21 | 33.43 | 20.90 | 47.82 | 7.12 | 5.63 |
| *Sequence Model* | 16.86 | 36.03 | 27.45 | 54.20 | 8.58 | 12.87 |
| SP4LP | **17.27** | **38.52** | **41.08** | **66.28** | **10.87** | **23.01** |

isolates the contribution of the sequential model in capturing relational patterns based solely on node features and structural path information. **(2) GNN + Shortest Path Length:** in this variant, the sequential model is completely removed. Link prediction is performed using only the learned node representations from the GNN, combined with the length of the shortest path between the target nodes. This evaluates the effectiveness of combining node embeddings with simple distance information, without explicitly modeling the intermediate nodes along the path.

Table 3 reports the results of these ablations, conducted on the Cora, Citeseer, and Pubmed datasets. Both variants show a clear performance drop compared to the full model, demonstrating the importance of jointly leveraging node representations and sequential model. The Sequence Model Only variant achieves reasonable results on simpler datasets such as Cora and Citeseer, but its performance degrades significantly on more complex datasets like Pubmed, highlighting the limitations of relying solely on raw node features without learned representations. The GNN + Shortest Path Length variant consistently underperforms across all datasets, indicating that simple distance information is insufficient. Indeed, replacing the embedding of the path with its length discards the information encoded in the node representations along the path. In contrast, when node embeddings are computed via message passing, they incorporate information from each node's local neighborhood, thus implicitly encoding a broader subgraph around the path, not just the path itself. Overall, the full model SP4LP achieves the best results across all datasets, confirming the importance of combining learned node representations with sequential modeling over the shortest paths to capture both local and global structural patterns in the graph.

## 5.3 SCALABILITY ANALYSIS

We assess the scalability of SP4LP by examining how its GPU memory consumption and inference time evolve as the batch size increases, in comparison to several baseline methods. The results are presented in Figure 3. In terms of GPU memory usage, SP4LP exhibits remarkable efficiency: memory consumption remains nearly constant across small to medium batch sizes, and increases moderately only for the largest batches, starting with 0.77 GB and reaching at most 6.84 GB while consistently avoiding out-of-memory (OOM) failures.

PEG also maintains low memory usage; however, this advantage is undermined by its impractically slow inference, limiting its applicability in large-scale scenarios. SEAL, while competitive with SP4LP in terms of inference speed, suffers from excessive memory consumption, rapidly exceeding 18 GB and encountering OOM issues beyond a batch size of 32,768. NCNC is even more

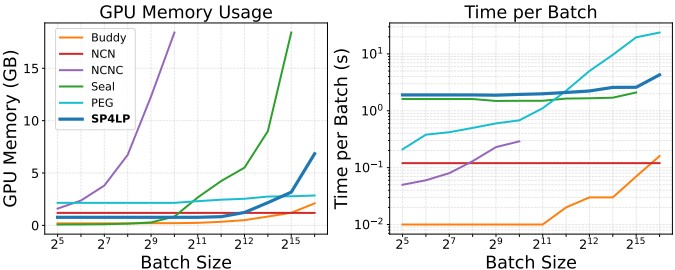

Figure 3: Inference time and GPU memory usage on ogbl-collab, measured during the prediction of a single batch of test links.

constrained, experiencing OOM failures already at relatively small batch sizes. Considering inference time, SP4LP matches the efficiency of SEAL, requiring only 2.1 seconds for o batch size of 16,384 and scaling smoothly to 4.29 seconds at 65,536. PEG, by contrast, is significantly slower, already taking 5 seconds at a batch size of 8,192 and exceeding 23 seconds at the maximum batch size tested. Although NCN and Buddy consistently achieve low inference times, this comes at the cost of substantially lower predictive performance, as they rely on simple heuristics with limited modeling capacity. This trade-off is clearly reflected in the results of Table 2. In summary, SP4LP achieves an excellent balance between low memory consumption, fast inference, high predictive accuracy, and strong model expressiveness.

## 6 CONCLUSION

We introduced SP4LP, a novel message-passing based framework for link representation that enhances the expressiveness of standard GNNs by incorporating sequential modeling over the shortest path between target nodes. SP4LP follows the GNN-then-SF paradigm, thus effectively combining the benefits of computing node embeddings only once with high expressive power. SP4LP explicitly models multi-hop relational patterns through the use of a sequence encoder applied to the node embeddings along the shortest path. We formally proved that SP4LP is strictly more expressive than several state-of-the-art link representation models. Extensive experiments under the HeaRT evaluation protocol confirm that SP4LP achieves state-of-the-art performance across diverse datasets.

As future work, we plan to extend SP4LP in several directions. First, we aim to incorporate multiple (or all) shortest paths between node pairs to improve robustness and capture richer structural signals. Second, we intend to explore the application of SP4LP to heterogeneous graphs: since shortest paths are well-defined in multi-relational settings and the path encoder can incorporate node and edge type embeddings, SP4LP can be adapted to heterogeneous link prediction with minimal architectural changes. Third, for very large graphs, we plan to investigate approximate shortest-path extraction techniques (e.g., truncated BFS, landmark-based methods) to further reduce preprocessing cost while retaining the structural benefits of path-based modeling.

## USE OF LARGE LANGUAGE MODELS (LLMS)

Large Language Models were employed to enhance the readability of the manuscript, refine the phrasing of selected passages, and provide assistance in code debugging. All original content was produced by the authors; LLMs were used exclusively to improve clarity and presentation.

## ETHICS STATEMENT

Our study does not involve human subjects or personally identifiable data. The datasets used are publicly available benchmarks or synthetically generated. We follow the ICLR Code of Ethics and note that our work raises no foreseeable ethical concerns beyond those inherent to the general study of machine learning with missing data.

## REPRODUCIBILITY STATEMENT

We have made every effort to ensure reproducibility. Details of the experimental setup are provided in Section 5, with dataset descriptions in Appendix C and complete training configurations in Appendix D. All proofs are included in Appendix A. Anonymous source code to reproduce our experiments is provided in the supplementary material.

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

# A PROOFS

**Proposition 4.4** SP4LP does not suffer from the automorphic node problem.

*Proof.* According to Proposition 2.9, a model $M$ suffers from the *automorphic node problem* if, for any graph $G = (V, E, \mathbf{X}^0)$, for any pairs of links $(u, v), (u', v') \in V \times V$, for any $F \in M$ it holds that:

$$(u, v) \not\simeq (u', v'), \quad u \simeq u', \quad v \simeq v', \quad \text{and} \quad F((u, v), G) \neq F((u', v'), G).$$

In order to prove that SP4LP does not suffer from the automorphic node problem, it suffices to provide an example of a graph $G = (V, E, X^0)$ and node pairs $(u, v), (u', v') \in V \times V$ such that:

$$(u, v) \not\simeq (u', v'), \quad u \simeq u', \quad v \simeq v', \quad \text{and} \quad \text{SP4LP}((u, v), G) \neq \text{SP4LP}((u', v'), G).$$

Such an example is provided in Figure 1: the shortest path from $v$ to $u'$ consists of $v$, an orange node, and $u'$, whereas the shortest path from $v$ to $u$ includes $v$, an orange node, a yellow node, another orange node, and finally $u$. Thus, simply using a summation as function $\phi$, leads to distinct representations for links $(v, u)$ and $(v, u')$. □

**Theorem 4.5** SP4LP is strictly more expressive than Pure GNN, NCN, BUDDY, NBFnet and Neo-GNN.

*Proof.* We proceed by prove each comparison separately.

- SP4LP is strictly more expressive than Pure GNN.

  We prove this by noting that SP4LP architecture (Equation 8) generalizes that of pure GNNs, meaning that SP4LP can simulate any pure GNN by simply ignoring the shortest path information. Thus, for any pair of non-automorphic links that a specific pure GNN can distinguish, there exists a configuration of SP4LP that distinguishes them as well. We can conclude that SP4LP is strictly more expressive than pure GNNs considering as example of pair of links indistinguishable by GNNs but distinguishable by SP4LP the one provided in the proof of Proposition 4.4.

- SP4LP is strictly more expressive than NCN.

  We prove this in two steps: (1) When two links share no common neighbors, NCN on them reduces to a pure GNN. As we have already proved that SP4LP is strictly more expressive than pure GNNs, it follows that SP4LP is also strictly more expressive than NCN in this case. (2) When the links have common neighbors, setting AGG as summation, $\rho$ as the Hadamard product between the endpoint representations and the concatenation with the result of the aggregation, and $\phi$ as the identity function, SP4LP reduces exactly to NCN. Therefore, if NCN can distinguish two links under some configuration, SP4LP can as well. By definition 2.10), this implies that SP4LP is more expressive than NCN. We can conclude that SP4LP is strictly more expressive than NCN considering the example in Figure 4. The graph is regular and thus all nodes receive the same embedding from a GNN. Consider nodes $u$ and $v$: $N(u) \cap N(v) = N(u) \cap N(v') = \emptyset$. In this case, NCN reduces to a pure GNN and is thus unable to distinguish the links $(u, v)$ and $(u', v')$. Now, let $\mathbf{x} \in \mathbb{R}^d$ be the representation assigned to every node by a GNN. Then, the representation of the shortest path $\mathcal{P}_G^*(u, v)$ is simply $(\mathbf{x}, \mathbf{x}, \mathbf{x})$, while $\mathcal{P}_G^*(u', v') = (\mathbf{x}, \mathbf{x}, \mathbf{x}, \mathbf{x})$. Even using a simple sum as aggregation function, SP4LP successfully distinguishes between the two links.

- SP4LP is strictly more expressive than Neo-GNN and BUDDY.

  We have already shown that SP4LP is strictly more expressive than NCN. In Theorem 2 of the NCN paper Wang et al. (2024), it has been proved that NCN is more expressive than both Neo-GNN and BUDDY. It follows that SP4LP is strictly more expressive than Neo-GNN and BUDDY as well.

- SP4LP is strictly more expressive than NBFNet.

  We prove that NBFNet is as expressive as a pure GNN. Therefore, since we have already proven that SP4LP is strictly more expressive than pure GNNs, it immediately follows that SP4LP is also strictly more expressive than NBFNet.

  To complete the argument, we prove that NBFNet is as expressive as a pure GNN. First of all we report the formulation of NBFNet for simple undirected graph. Given a graph $G = (V, E, \mathbf{X}^0)$, NBFNet assigns a representation $\mathbf{x}(u, v)$ to each edge $(u, v) \in E$. The iterative update rule follows a message-passing scheme:

  $$\begin{aligned} \mathbf{x}_{(u,v)}^{(0)} &= \text{INDICATOR}(\mathbf{x}_u^0, \mathbf{x}_v^0), \\ \mathbf{x}_{(u,v)}^{(l)} &= \text{AGGREGATE}\left( \left\{ \text{MESSAGE}(h_{(i,j)}^{(l-1)}) \mid (i,j) \in N(u,v) \right\} \cup \{h_{(u,v)}^{(0)}\} \right) \end{aligned} \tag{9}$$

  where INDICATOR assigns an initial representation based on the nodes $u, v \in V$ and $N(u, v)$ is the set of edges incident to $(u, v)$.

  We prove that, at any layer $l$, the representations of the two links produced by a pure GNN are equal if and only if also the ones produced by NBFNet are equal, i.e.,

  $$\mathbf{x}_{(u,v)}^{\text{GNN}^l} = \mathbf{x}_{(i,j)}^{\text{GNN}^l} \Leftrightarrow \mathbf{x}_{(u,v)}^{\text{NBF}^l} = \mathbf{x}_{(i,j)}^{\text{NBF}^l} \quad \forall l \tag{10}$$

  where $\mathbf{x}_{(u,v)}^{\text{NBF}^l}$ and $\mathbf{x}_{(i,j)}^{\text{NBF}^l}$ are calculated via Equation 9, while $\mathbf{x}_{(u,v)}^{\text{GNN}^l}$ and $\mathbf{x}_{(i,j)}^{\text{GNN}^l}$ are calculated following the standard message passing scheme reported below:

  $$\begin{aligned} \mathbf{x}_v^{\text{GNN}^0} &= \mathbf{x}_v^0, \\ \mathbf{x}_v^{\text{GNN}^l} &= \text{COMB}\left( \mathbf{x}_v^{\text{GNN}^{l-1}}, \text{AGG}\left( \{\mathbf{x}_u^{\text{GNN}^{l-1}} \mid u \in N(v)\} \right) \right) \\ \mathbf{x}_{(u,v)}^{\text{GNN}^l} &= g(\mathbf{x}_u^{\text{GNN}^l}, \mathbf{x}_v^{\text{GNN}^l}) \end{aligned} \tag{11}$$

  Let the functions INDICATOR, AGGREGATE and MESSAGE of Equation 9, as well as the functions COMB, AGG and $g$ be injective. We prove Equation 10 by induction on the number of layer $l$.

  **Base Case:** $l = 0$

  $$\mathbf{x}_{(u,v)}^{\text{GNN}^0} = \mathbf{x}_{(i,j)}^{\text{GNN}^0} \overset{(11)}{\Longleftrightarrow} g(\mathbf{x}_u^0, \mathbf{x}_v^0) = g(\mathbf{x}_i^0, \mathbf{x}_j^0) \overset{\text{inj}}{\Longleftrightarrow} (\mathbf{x}_u^0, \mathbf{x}_v^0) = (\mathbf{x}_i^0, \mathbf{x}_j^0) \overset{\text{inj}}{\Longleftrightarrow}$$

  $$\overset{\text{inj}}{\Longleftrightarrow} \text{INDICATOR}(\mathbf{x}_u^0, \mathbf{x}_v^0) = \text{INDICATOR}(\mathbf{x}_i^0, \mathbf{x}_j^0) \overset{(9)}{\Longleftrightarrow} \mathbf{x}_{(u,v)}^{\text{NBF}^0} = \mathbf{x}_{(i,j)}^{\text{NBF}^0}$$

  **Inductive Step** We assume Equation 10 holds for $l - 1$ and prove it holds for $l$.

  In particular, we want to prove

  $$\mathbf{x}_{(u,v)}^{\text{GNN}^l} = \mathbf{x}_{(i,j)}^{\text{GNN}^l} \Longleftrightarrow \mathbf{x}_{(u,v)}^{\text{NBF}^l} = \mathbf{x}_{(i,j)}^{\text{NBF}^l} \tag{12}$$

  using the inductive hypothesis

  $$\mathbf{x}_{(u,v)}^{\text{GNN}^{l-1}} = \mathbf{x}_{(i,j)}^{\text{GNN}^{l-1}} \Longleftrightarrow \mathbf{x}_{(u,v)}^{\text{NBF}^{l-1}} = \mathbf{x}_{(i,j)}^{\text{NBF}^{l-1}} \tag{13}$$

  Applying Equation 11 to the left-hand side of Equation 12 we get

  $$\mathbf{x}_{(u,v)}^{\text{GNN}^l} = \mathbf{x}_{(i,j)}^{\text{GNN}^l}$$

  $$\overset{(11)}{\Longleftrightarrow}$$

  $$g(\text{COMB}(\mathbf{x}_u^{\text{GNN}^{l-1}}, \text{AGG}(\{\mathbf{x}_x^{\text{GNN}^{l-1}} \mid x \in N(u)\})), \text{COMB}(\mathbf{x}_v^{\text{GNN}^{l-1}}, \text{AGG}(\{\mathbf{x}_y^{\text{GNN}^{l-1}} \mid y \in N(u)\})))$$

  $$=$$

  $$g(\text{COMB}(\mathbf{x}_i^{\text{GNN}^{l-1}}, \text{AGG}(\{\mathbf{x}_m^{\text{GNN}^{l-1}} \mid m \in N(i)\})), \text{COMB}(\mathbf{x}_j^{\text{GNN}^{l-1}}, \text{AGG}(\{\mathbf{x}_n^{\text{GNN}^{l-1}} \mid n \in N(j)\}))).$$

Given the injectivity of $g$, COMB and AGG, this is equivalent to

$$\mathbf{x}_u^{\text{GNN}^{l-1}} = \mathbf{x}_i^{\text{GNN}^{l-1}} \wedge \{\mathbf{x}_x^{\text{GNN}^{l-1}} \mid x \in N(u)\} = \{\mathbf{x}_m^{\text{GNN}^{l-1}} \mid m \in N(i)\} \wedge$$
$$\wedge \mathbf{x}_v^{\text{GNN}^{l-1}} = \mathbf{x}_j^{\text{GNN}^{l-1}} \wedge \{\mathbf{x}_y^{\text{GNN}^{l-1}} \mid y \in N(v)\} = \{\mathbf{x}_n^{\text{GNN}^{l-1}} \mid n \in N(j)\}$$
$$\Longleftrightarrow$$
$$\{(\mathbf{x}_u^{\text{GNN}^{l-1}}, \mathbf{x}_x^{\text{GNN}^{l-1}}) \mid x \in N(u)\} = \{(\mathbf{x}_i^{\text{GNN}^{l-1}}, \mathbf{x}_m^{\text{GNN}^{l-1}}) \mid m \in N(i)\}$$
$$\wedge$$
$$\{(\mathbf{x}_v^{\text{GNN}^{l-1}}, \mathbf{x}_y^{\text{GNN}^{l-1}}) \mid y \in N(v)\} = \{(\mathbf{x}_j^{\text{GNN}^{l-1}}, \mathbf{x}_n^{\text{GNN}^{l-1}}) \mid n \in N(j)\}$$
$$\Longleftrightarrow$$
$$\{(\mathbf{x}_u^{\text{GNN}^{l-1}}, \mathbf{x}_x^{\text{GNN}^{l-1}}) \mid x \in N(u)\} \cup \{(\mathbf{x}_v^{\text{GNN}^{l-1}}, \mathbf{x}_y^{\text{GNN}^{l-1}}) \mid y \in N(v)\}$$
$$=$$
$$\{(\mathbf{x}_i^{\text{GNN}^{l-1}}, \mathbf{x}_m^{\text{GNN}^{l-1}}) \mid m \in N(i)\} \cup \{(\mathbf{x}_j^{\text{GNN}^{l-1}}, \mathbf{x}_n^{\text{GNN}^{l-1}}) \mid n \in N(j)\}.$$

By Definition of $N(u,v)$ (Equation 9), this is equivalent to

$$\{(\mathbf{x}_w^{\text{GNN}^{l-1}}, \mathbf{x}_t^{\text{GNN}^{l-1}}) \mid (w,t) \in N(u,v)\} = \{(\mathbf{x}_a^{\text{GNN}^{l-1}}, \mathbf{x}_b^{\text{GNN}^{l-1}}) \mid (a,b) \in N(i,j)\}$$
$$\overset{\text{inj}}{\Longleftrightarrow}$$
$$\{g(\mathbf{x}_w^{\text{GNN}^{l-1}}, \mathbf{x}_t^{\text{GNN}^{l-1}}) \mid (w,t) \in N(u,v)\} = \{g(\mathbf{x}_a^{\text{GNN}^{l-1}}, \mathbf{x}_b^{\text{GNN}^{l-1}}) \mid (a,b) \in N(i,j)\}$$
$$\overset{(11)}{\Longleftrightarrow}$$
$$\{\mathbf{x}_{(w,t)}^{\text{GNN}^{l-1}} \mid (w,t) \in N(u,v)\} = \{\mathbf{x}_{(a,b)}^{\text{GNN}^{l-1}} \mid (a,b) \in N(i,j)\}$$
$$\overset{\text{IND. HYP.}(13)}{\Longleftrightarrow}$$
$$\{\mathbf{x}_{(w,t)}^{\text{NBF}^{l-1}} \mid (w,t) \in N(u,v)\} = \{\mathbf{x}_{(a,b)}^{\text{NBF}^{l-1}} \mid (a,b) \in N(i,j)\}.$$

Using the hypotheses of injective AGG and MESSAGE, this is equivalent to:

$$\text{AGG}(\{\text{MESSAGE}(\mathbf{x}_{(w,t)}^{\text{NBF}^{l-1}}) \mid (w,t) \in N(u,v)\}) = \text{AGG}(\{\text{MESSAGE}(\mathbf{x}_{(a,b)}^{\text{NBF}^{l-1}}) \mid (a,b) \in N(i,j)\})$$
$$\overset{(9)}{\Longleftrightarrow}$$
$$\mathbf{x}_{(u,v)}^{\text{NBF}^l} = \mathbf{x}_{(i,j)}^{\text{NBF}^l} \tag{14}$$

which complete the proof.

$\square$

# B  ADDITIONAL RESULTS

We complement the main results of Section 5 with additional tables reporting the standard deviation computed over five different random seeds, to better assess the stability of each method.

**Real-world Datasets: Results with Standard Deviations**    Table 4 and Table 5 expand the main results in Table 2 by including both the mean and standard deviation of MRR and Hits@$K$ across runs.

**Ablation Study: Results with Standard Deviations**    Similarly, Table 6 complements the ablation results in Table 3 by reporting mean and standard deviation for Cora, Citeseer, and Pubmed.

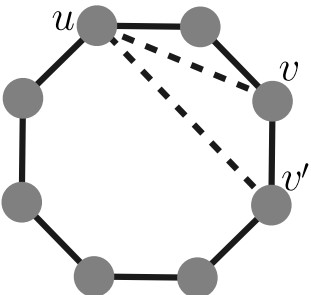

Figure 4: Links $(u, v), (u, v')$ are not distinguished by NCN while are distinguished by SP4LP.

| Models | Cora MRR | Citeseer MRR | Pubmed MRR | Ogbl-ddi MRR | Ogbl-collab MRR | Ogbl-ppa MRR | Ogbl-Citation2 MRR |
|---|---|---|---|---|---|---|---|
| Node2Vec | 14.47 (± 0.60) | 21.17 (± 1.01) | 3.94 (± 0.24) | 11.14 (± 0.95) | 4.68 (± 0.08) | 18.33 (± 0.10) | 14.67 (± 0.18) |
| MF | 6.20 (± 1.42) | 7.80 (± 0.79) | 4.46 (± 0.32) | 13.99 (± 0.47) | 4.89 (± 0.25) | 22.47 (± 1.53) | 8.72 (± 2.60) |
| MLP | 13.52 (± 0.65) | 22.62 (± 0.55) | 6.41 (± 0.25) | N/A | 5.37 (± 0.14) | 0.98 (± 0.00) | 16.32 (± 0.07) |
| GCN | 16.61 (± 0.30) | 21.09 (± 0.88) | 7.13 (± 0.27) | 13.46 (± 0.34) | 6.09 (± 0.38) | 26.94 (± 0.48) | 19.98 (± 0.35) |
| GAT | 13.84 (± 0.68) | 19.58 (± 0.84) | 4.95 (± 0.14) | 12.92 (± 0.39) | 4.18 (± 0.33) | OOM | OOM |
| SAGE | 14.74 (± 0.69) | 21.09 (± 1.15) | 9.40 (± 0.70) | 12.60 (± 0.72) | 5.53 (± 0.50) | 27.27 (± 0.30) | 22.05 (± 0.12) |
| GAE | 18.32 (± 0.41) | 25.25 (± 0.82) | 5.27 (± 0.25) | 3.49 (± 1.73) | OOM | OOM | OOM |
| SEAL | 10.67 (± 3.46) | 13.16 (± 1.66) | 5.88 (± 0.53) | 9.99 (± 0.90) | 6.43 (± 0.32) | 29.71 (± 0.71) | 20.60 (± 1.28) |
| BUDDY | 13.71 (± 0.59) | 22.84 (± 0.36) | 7.56 (± 0.18) | 12.43 (± 0.50) | 5.67 (± 0.36) | 27.70 (± 0.33) | 19.17 (± 0.10) |
| Neo-GNN | 13.95 (± 0.39) | 17.34 (± 0.84) | 7.74 (± 0.30) | 10.86 (± 2.16) | 5.23 (± 0.90) | 21.68 (± 1.14) | 16.12 (± 0.25) |
| NCN | 14.66 (± 0.95) | 28.65 (± 1.21) | 5.84 (± 0.22) | 12.86 (± 0.78) | 5.09 (± 0.38) | 35.06 (± 0.26) | 23.35 (± 0.28) |
| NCNC | 14.98 (± 1.00) | 24.10 (± 0.65) | 8.58 (± 0.59) | >24h | 4.73 (± 0.86) | 33.52 (± 0.26) | 19.61 (± 0.54) |
| NBFNet | 13.56 (± 0.58) | 14.29 (± 0.80) | >24h | >24h | OOM | OOM | OOM |
| PEG | 15.73 (± 0.39) | 21.01 (± 0.77) | 4.40 (± 0.41) | 12.05 (± 1.14) | 4.83 (± 0.21) | OOM | OOM |
| LPFORMER | 16.80 (± 0.52) | 26.34 (± 0.67) | 9.99 (± 0.52) | 13.20 (± 0.54) | 7.62 (± 0.26) | 40.25 (± 0.24) | 24.70 (± 0.55) |
| SP4LP | 17.27 (± 0.57) | 41.08 (± 1.84) | 10.87 (± 0.31) | 15.00 (± 0.57) | 9.46 (± 0.55) | 36.45 (± 1.21) | 24.91 (± 0.41) |

Table 4: MRR results across all datasets, following the HeaRT evaluation setting Li et al. (2023). The top three results for each metric are highlighted using **first**, *second*, and **third**. *OOM* indicates that the model ran out of memory, while *>24h* denotes that the method did not complete within 24 hours.

## C   DATASETS STATISTICS

Table 7 summarizes the main datasets used in our link prediction experiments. Cora, Citeseer, and Pubmed are well-known citation networks frequently used as benchmarks for graph-based learning methods. These datasets are relatively small, both in the number of nodes and edges. In contrast, the datasets from the Open Graph Benchmark (OGB), namely ogbl-collab and ogb-ddi, are substantially larger and more complex, offering challenging scenarios for evaluating model scalability and performance on large-scale graphs.

For all datasets, we adopt the splits provided by the Li et al. (2023) setting.

## D   EXPERIMENTAL SETTINGS

This section outlines the experimental setup used to evaluate all models. We describe the computational resources and the hyperparameter search space. Moreover for SP4LP we include details regarding how the calculation of the shortest path is performed. Details are reported below.

**Computational Resources**   All experiments were conducted on a workstation running Ubuntu 22.04 with an AMD Ryzen 9 7950X CPU (32 threads), 124GB of RAM, and two NVIDIA GeForce RTX 4090 GPUs (24GB each).

**Hyperparameters**   All models are tuned using a grid search over learning rate $\in [1 \times 10^{-2}, 1 \times 10^{-3}]$, dropout $\in [0, 0.7]$, weight decay $\in [0, 10^{-4}, 10^{-7}]$, number of GNN layers $\in \{1, 2, 3\}$, hid-

| Models | Cora
Hits@10 | Citeseer
Hits@10 | Pubmed
Hits@10 | Ogbl-ddi
Hits@20 | Ogbl-collab
Hits@20 | Ogbl-ppa
Hits@20 | Ogbl-Citation2
Hits@20 |
|---|---|---|---|---|---|---|---|
| Node2Vec | 32.77 (± 1.29) | 45.82 (± 2.01) | 8.51 (± 0.77) | 63.63 (± 2.05) | 16.84 (± 0.17) | 53.42 (± 0.11) | 42.68 (± 0.20) |
| MF | 15.26 (± 3.39) | 16.72 (± 1.99) | 9.42 (± 0.80) | 59.50 (± 1.68) | 18.86 (± 0.40) | 70.71 (± 4.82) | 29.64 (± 7.30) |
| MLP | 31.01 (± 1.71) | 48.02 (± 1.79) | 15.04 (± 0.67) | N/A | 16.15 (± 0.27) | 1.47 (± 0.00) | 43.15 (± 0.10) |
| GCN | 36.26 (± 1.14) | 47.23 (± 1.88) | 15.22 (± 0.57) | 64.76 (± 1.45) | **22.48** (± 0.81) | 68.38 (± 0.73) | 51.72 (± 0.46) |
| GAT | 32.89 (± 1.27) | 45.30 (± 1.30) | 9.99 (± 0.64) | *66.83* (± 2.23) | 18.30 (± 1.42) | OOM | OOM |
| SAGE | 34.65 (± 1.47) | 48.75 (± 1.85) | *20.54* (± 1.40) | **67.19** (± 1.18) | 21.26 (± 1.32) | 69.49 (± 0.43) | 53.13 (± 0.15) |
| GAE | *37.95* (± 1.24) | 49.65 (± 1.48) | 10.50 (± 0.46) | 17.81 (± 9.80) | OOM | OOM | OOM |
| SEAL | 24.27 (± 6.74) | 27.37 (± 3.20) | 12.47 (± 1.23) | 49.74 (± 2.39) | 21.57 (± 0.38) | 76.77 (± 0.94) | 48.62 (± 1.93) |
| BUDDY | 30.40 (± 1.18) | 48.35 (± 1.18) | 16.78 (± 0.53) | 58.71 (± 1.63) | **23.35** (± 0.73) | 71.50 (± 0.68) | 47.81 (± 0.37) |
| Neo-GNN | 31.27 (± 0.72) | 41.74 (± 1.18) | 17.88 (± 0.71) | 51.94 (± 10.33) | 21.03 (± 3.39) | 64.81 (± 2.26) | 43.17 (± 0.53) |
| NCN | 35.14 (± 1.04) | **53.41** (± 1.46) | 13.22 (± 0.56) | *65.82* (± 2.66) | 20.84 (± 1.31) | **81.89** (± 0.31) | *53.76* (± 0.20) |
| NCNC | *36.70* (± 1.57) | *53.72* (± 0.97) | *18.81* (± 1.16) | >24h | 20.49 (± 3.97) | **82.24** (± 0.40) | 51.69 (± 1.48) |
| NBFNet | 31.12 (± 0.75) | 31.39 (± 1.34) | >24h | >24h | OOM | OOM | OOM |
| PEG | 36.03 (± 0.75) | 45.56 (± 1.38) | 8.70 (± 1.26) | 50.12 (± 6.55) | 18.29 (± 1.06) | OOM | OOM |
| LPFORMER | 33.27 (± 1.33) | 51.58 (± 1.83) | 22.71 (± 1.30) | 35.23 (± 0.37) | *23.05* (± 0.16) | **84.01** (± 0.10) | *57.30* (± 0.50) |
| SP4LP | **38.52** (± 1.19) | **66.28** (± 0.63) | **23.01** (± 0.39) | 47.96 (± 3.82) | 20.00 (± 1.20) | 76.9 (± 1.11) | 55.45 (± 0.92) |

Table 5: Hits@K (%) results across all datasets, following the HeaRT evaluation setting Li et al. (2023). The top three results for each metric are highlighted using **first**, *second*, and **third**. *OOM* indicates that the model ran out of memory, while *>24h* denotes that the method did not complete within 24 hours.

| Models | Cora | | Citeseer | | Pubmed | |
|---|---|---|---|---|---|---|
| | MRR | Hits@10 | MRR | Hits@10 | MRR | Hits@10 |
| *GNN + SP len.* | 14.21 (± 1.44) | 33.43 (± 2.69) | 20.90 (± 0.79) | 47.82 (± 1.11) | 7.12 (± 0.41) | 5.63 (± 0.52) |
| *Sequence Model* | 16.86 (± 1.26) | 36.03 (± 1.75) | 27.45 (± 1.55) | 54.20 (± 2.35) | 8.58 (± 0.75) | 12.87 (± 0.85) |
| SP4LP | **17.27** (± 0.57) | **38.52** (± 1.19) | **41.08** (± 1.84) | **66.28** (± 0.63) | **10.87** (± 0.31) | **23.01** (± 0.39) |

Table 6: Ablation study results (%). MRR and Hits@K with mean and std. deviations over 5 runs with different seeds.

den dimensions $\in \{32, 64, 128, 256\}$ and prediction layers $\in \{1, 2, 3\}$. For large-scale datasets, we follow the reduced search space adopted in Li et al. (2023) to avoid excessive compute. For SP4LP, we additionally explore the choice of GNN component $\in \{\text{GCN}, \text{GraphSAGE}, \text{GAT}\}$ and sequence model $\in \{\text{LSTM}, \text{Transformer}\}$, the best models are shown in Table 8. The best hyperparameters are selected based on validation performance. All reported metrics are averaged over 5 different seeds.

**Shortest path calculation for SP4LP** To compute shortest paths for SP4LP, we use the networkx.shortest_path BFS implementation (Hagberg et al., 2008). Importantly, we compute and cache shortest paths only for the train/validation/test links provided by the HeaRT benchmark splits. If multiple links share an endpoint, only one BFS is required for that source node. This keeps preprocessing time low: in all datasets, the total number of BFS sources is small, and the full preprocessing step completes in seconds to a few minutes. The cached paths are reused across all training epochs and inference batches. No all-pairs shortest path computation is performed.

In our implementation of SP4LP, the sequential encoder can be instantiated as a Transformer with learnable positional encodings. We use PyTorch's TransformerEncoder, composed of TransformerEncoderLayer blocks with 4-head self-attention, feedforward sublayers, ReLU activation, and dropout. We tune the number of layers $\in \{1, 2\}$, attention heads $\in \{2, 3, 4\}$, and feedforward dimensionality $\in \{32, 64, 128\}$. A trainable positional embedding matrix is added to node embeddings to preserve path order. Variable-length paths are handled via a source key padding mask, and the output is aggregated using masked mean pooling followed by layer normalization. The resulting path representation is then combined with the GNN-derived embeddings of the source and target nodes to compute the final link score.

For the LSTM-based encoder, we implement both unidirectional and bidirectional variants using PyTorch's nn.LSTM. Input sequences are packed with pack_padded_sequence to handle variable-length

| Dataset | Cora | Citeseer | Pubmed | ogbl-collab | ogbl-ddi | ogbl-ppa | ogbl-citation2 |
|---------|------|----------|--------|-------------|----------|----------|----------------|
| #Nodes | 2708 | 3327 | 18717 | 235868 | 4267 | 576289 | 2927963 |
| #Edges | 5278 | 4676 | 44327 | 1285465 | 1334889 | 30326273 | 30561187 |

Table 7: Dataset statistics.

| Dataset | GNN model | Sequence model |
|---------|-----------|----------------|
| Cora | GCN | Transformer |
| Citeseer | GCN | LSTM |
| Pubmed | SAGE | Transformer |
| ogbl-collab | SAGE | Transformer |
| ogbl-ddi | GCN | Transformer |

Table 8: Best GNN and sequence models selected via hyperparameter tuning.

paths. We tune the number of layers $\in \{1, 2\}$ and hidden size $\in \{32, 64, 128\}$. The final hidden state (or the concatenation of forward and backward states in the bidirectional case) is used as the path representation and combined with the GNN-based node embeddings for link prediction. Node embeddings are obtained via a GNN encoder selected from GCN, GAT, or GraphSAGE, depending on the experimental setting.

# E COMPUTATIONAL COMPLEXITY

In this section, we analyze the computational complexity of our proposed SP4LP model for link prediction, following the formalism and notation introduced in Wang et al. (2024). This framework allows a direct comparison with prior approaches such as GAE, Neo-GNN, BUDDY, PEG, SEAL, NCN, and NCNC.

Let $n$ be the number of nodes, $m$ the number of edges, $\Delta$ the maximum degree, $d$ the dimensionality of node features or embeddings, $t$ the number of target links to predict, and $k$ the length of the shortest path between node pairs (typically $k \ll n$). We express total time complexity as $\mathcal{O}(B + C \cdot t)$, where $B$ is the precomputation cost (independent of $t$), and $C$ is the per-link cost.

Our SP4LP method follows the GNN-then-Structural-Feature (SF) paradigm Wang et al. (2024), applying a Message Passing Neural Network once across the entire graph to compute node embeddings, then leveraging shortest path extraction combined with sequence modeling for each candidate link.

Following the paradigm, the precomputation cost of SP4LP is:

$$B = \mathcal{O}(n\Delta d + nd^2 + T_{\text{sp}})$$

where: $n\Delta d$ accounts for neighborhood aggregation in the GNN, $nd^2$ represents linear transformations in GNN layers, $T_{\text{sp}}$ is the cost of computing shortest paths (e.g., via BFS).

For sparse graphs, single-source BFS runs in $O(m)$ time. Although all-pairs shortest paths (APSP) would cost $O(nm)$, SP4LP never computes APSP. In all benchmark settings, we compute shortest paths only for the supervised train/validation/test links. Let $S$ be the number of distinct endpoint nodes among these links. The total preprocessing cost is therefore

$$T_{\text{sp}} = O(S \cdot m),$$

typically several orders of magnitude smaller than $O(nm)$. Once these paths are cached, no BFS is performed during training or batched inference. In fully inductive scenarios, shortest-path trees for new nodes can be obtained via a single BFS per new node, which provides distances and parent pointers to all existing nodes at once. The per-link inference cost of SP4LP is thus

$$C = O(kd + T_{\text{seq}}),$$

where $k$ is the path length and $d$ the embedding dimension, making it independent of the graph size $n$. $T_{\text{seq}}$ denotes the cost of a sequence model (e.g., LSTM, Transformer) applied to the node embeddings along that path.

Importantly, since both the GNN embeddings and shortest paths can be precomputed, $B$ is incurred only once. Given that $k$ is typically small in practice, $C$ remains low even at scale. Unlike subgraph-based methods such as SEAL, which require per-link GNN inference over extracted neighborhoods, SP4LP performs lightweight sequence modeling on compact paths, supporting efficient batched inference. Additionally, the framework is flexible: $T_{\text{sp}}$ can be cached or approximated Sommer (2014), and $T_{\text{seq}}$ depends on the chosen architecture and path length.

We summarize the complexities in the following table:

| Method | B | C |
|---|---|---|
| GAE | $n\Delta d + nd^2$ | $d^2$ |
| Neo-GNN | $n\Delta d + nd^2 + n\Delta^l$ | $\Delta^l + d^2$ |
| BUDDY | $n\Delta d + nh$ | $h + d^2$ |
| SEAL | $0$ | $\Delta^{(l+1)}d + \Delta^l d^2$ |
| NCN | $n\Delta d + nd^2$ | $\Delta d + d^2$ |
| NCNC | $n\Delta d + nd^2$ | $\Delta^2 d + \Delta d^2$ |
| PEG | $n\Delta d + nd^2 + nD^2$ | $d^2$ |
| **SP4LP** | $n\Delta d + nd^2 + T_{\text{sp}}$ | $kd + T_{\text{seq}}$ |

Table 9: Comparison of precomputation ($B$) and per-link ($C$) complexities across methods.

All methods conform to the form $\mathcal{O}(B + C \cdot t)$, with a one-time graph-level computation and a per-target-link cost. In the table, $h$ is the cost of the hash function used in BUDDY, $l$ denotes the radius of local neighborhoods in Neo-GNN and SEAL, and $D$ denotes the number of Laplacian eigenvectors used for spectral positional encoding in PEG. GAE, NCN, and BUDDY are efficient but limited in expressiveness. NCNC enhances NCN via soft neighbor completion at a slightly higher cost. PEG incorporates spectral encoding ($\mathcal{O}(nD^2)$), while SEAL is the most computationally intensive due to subgraph extraction and per-link GNN processing. SP4LP, in contrast, strikes a balance between efficiency and expressiveness by combining GNN-based embeddings with sequence modeling over shortest paths, whose cost can be mitigated via approximation techniques Sommer (2014), ensuring scalability for large graphs.

