# OpenReview forum: "Improving expressivity in Link Prediction with GNNs via the Shortest Path"
_ICLR.cc/2026/Conference — Submitted to ICLR 2026_

### Official Review · Reviewer_jKya · 2025-10-26

**Soundness:** 3
**Presentation:** 4
**Contribution:** 3
**Rating:** 4
**Confidence:** 3

**Summary:**

SP4LP computes node embeddings with a GNN, then encodes the shortest-path node sequence between endpoints to build a path-aware link representation that is provably more expressive and achieves SOTA on HeaRT benchmarks.

**Strengths:**

Sound theory;
Completed proofs;
The representation can include both feature and path information.

**Weaknesses:**

Is computing shortest paths too slow on large graphs?

 Path-based representations can be unstable: if any edge on a path is (re)predicted to exist, the chosen path may change and thus alter the representation. So the method is not isolated to downstream training. Even when using only the shortest path, the hop length is usually be capped to keep structure local and reduce variance and such unstability.

 If only the shortest path is used, paths may frequently pass through high-degree “popular” nodes, making many representations highly correlated. Would multiple paths or de-hubbed routing help?

 A shortest path may not encode semantic signals; in an unsupervised setting this adds uncertainty. It would help to discuss when the method is appropriate and to design pre-application diagnostics to evaluate data suitability.

 Using the representation to inject even a small amount of structural context is reasonable, but please clarify what could be lost by focusing on shortest paths (e.g., robustness vs. information loss, local vs. global structure) and quantify the trade-offs.

**Questions:**

- Should we see the robustness of the method with possible longer path been invovled? Even they are the shortest path.

---

> ### Author Response · Authors · 2025-11-18
>
> Thank you for the thoughtful review! We’re glad you found the **theory solid**, the **proofs complete**, and the ability to **combine features with path information valuable**. Below you can find our responses to each of your comments.
>
> ---
>
> **Shortest paths on large graphs**
> The computational cost of shortest-path extraction is already detailed in the paper: for a graph with nn nodes and mmedges, all-pairs shortest paths can be computed with multi-source BFS in O(n(n+m)), which reduces to O(n2) for sparse graphs. Importantly, this preprocessing is performed once, before training, and SP4LP only retrieves the precomputed sequences during training and inference.
> For very large graphs, we agree that exact shortest paths may become expensive; however, SP4LP naturally supports approximate alternatives such as truncated BFS and landmark-based methods, which reduce preprocessing cost while preserving path-based structural information. We now make this point explicit in the paper and mention approximate shortest-path extraction as a future direction.
>
> ---
>
> **Unstable path-based representations.** We thank the reviewer for raising this point. However, SP4LP **does not suffer from the instability described**. In our framework, the graph topology is never updated, no edges are (re)predicted during training, and all shortest paths are computed once on the fixed input graph. As a result, the path used for each link is deterministic and cannot fluctuate across training steps. The issue mentioned typically arises in methods where the graph is dynamically modified, which is not the case here. We also note that **existing structural-feature-based methods** such as SEAL, Buddy, NCN, NCNC, LPFormer, NBFNet, NeoGNN and PEG operate on a fixed graph and **do not update the topology** with predicted edges.
>
> ---
>
> **High-Degree Nodes.** In SP4LP, using a single shortest path does not lead to excessive correlation in practice because the GNN encodes local contextual structure, so even high-degree nodes have embeddings that vary across neighborhoods. Prior works [1,2] on shortest-path–based representations similarly find that a single shortest path provides a strong and sufficiently discriminative signal. We agree that using multiple paths or hub-attenuated routing is an interesting extension, and our framework can naturally accommodate it. We clarified this point at the end of the conclusion section.
>
> ---
>
> **Semantic signals.** Thank you for raising this point. While we agree that shortest paths may lack semantic signals in some graphs, we would like to clarify that this is not the typical case across many link prediction benchmarks. Numerous prior works show that geodesic structure often provides strong relational cues even in unsupervised settings, and \our further enriches this signal by applying a sequence encoder to contextualized node embeddings. This allows semantic patterns to emerge even when the underlying path is structurally simple.
>
> ---
>
> **Trade-offs and Limitations of Shortest-Path–Based Representations.** Thank you for this helpful comment. We now explicitly discuss what may be lost when relying on shortest-path structure, including robustness issues and potential information loss, and we summarize the local–global trade-off. This clarification has been added in Section 2.
>
>
> [1] Ying, Chengxuan, et al. "Do transformers really perform badly for graph representation?." Advances in neural information processing systems 34 (2021): 28877-28888.
> [2] Airale, Louis, et al. "Simple Path Structural Encoding for Graph Transformers." Forty-second International Conference on Machine Learning.

---

> > ### Author Response · Authors · 2025-11-24
> >
> > Dear Reviewer, this is a polite follow-up to inquire whether you have been able to review our reply to the points you raised.
> >
> > Thank you for your time and consideration.

---

> ### Author Response · Authors · 2025-11-26
>
> Dear Reviewer,
>
> thank you again for your thoughtful comments. If any additional clarification from our side could assist the evaluation, we are happy to provide it.
>
> Kind regards.

---

### Official Review · Reviewer_GvsM · 2025-10-29

**Soundness:** 3
**Presentation:** 3
**Contribution:** 3
**Rating:** 8
**Confidence:** 5

**Summary:**

The paper tackles a well‑known issue in link prediction with message‑passing GNNs: when we form a link embedding only from the two endpoint node embeddings, structurally different links can collapse to the same representation—especially when endpoints are automorphic. The authors introduce SP4LP, a GNN‑then‑SF framework that first runs a standard GNN to compute node embeddings and then forms a link representation by encoding the shortest path between the two endpoints as an ordered sequence with an LSTM/Transformer (or an injective sum) before combining it with the endpoints’ embeddings. On the theory side, the paper claims SP4LP is strictly more expressive than several popular baselines (Pure GNNs, NCN, BUDDY, NBFNet, Neo‑GNN). Empirically, the authors adopt the HeaRT evaluation protocol and report results on Planetoid and OGB datasets, which shows advantage performance of SP4LP.

**Strengths:**

Strengths:

[S1] This paper is an intuitive and natural extension from NCN --> NCNC, which consider not only 1-hop neighborhood overlap, but also high-order overlap.

[S2] Competitive results under a hard protocol, the HeaRT setting. SP4LP achieves top or near‑top performance on several datasets and is consistently strong across metrics.

[S3] The efficiency of the model is as good as other baselines, while excelling at the performance.

**Weaknesses:**

Weakness:

[W1] Several claims made by the paper is not rigourus.

[W1.1] For example, the paper said "NCNC depends on the presence of common neighbors between link endpoints". However, NCNC can complete (this is what the last "C" comes from NCNC) the link if that node is connected to one of the target endpoint.

[W1.2] The paper claims that "... prove that SP4LP is strictly more expressive than existing SF-and-GNN approaches". However, MPLP[1], a SF-and-GNN method, actually is **not less** expressive than SP4LP and NCN. Consider two rook's graph but with different degrees. These two graphs are strongly regular graphs. Any non-adjacent nodes on rook's graph will have two common neighbors. Then for any two non adjacent nodes, the final link representation of SP4LP or NCN on one rook's graph will be the same as the other, because they have no distinct node features or distinct shortest paths pattern. However, MPLP is able to distinguish because its norm rescaling can include node degree information. BTW, MPLP also have a good performance on HeaRT setting, which the authors may overlook.

[1] Pure Message Passing Can Estimate Common Neighbor for Link Prediction. Neurips 24'

**Questions:**

1. For the sequential modeling over the node embedding on the shortest path, is the positional embedding necessary?

---

> ### Author Response · Authors · 2025-11-18
>
> We thank the reviewer for the constructive and thoughtful feedback. We appreciate the positive remarks regarding the **intuition behind our approach** [S1], the **strong empirical performance** under the HeaRT protocol [S2], and the **competitive efficiency** of our model [S3]. Below, we address each of your comments in detail and clarify the points you raised. All corresponding modifications in the revised manuscript are highlighted in blue to facilitate verification.
>
> ---
>
> **W1.1.** You are correct: NCNC can complete second-order neighbours even when direct common neighbours are absent. **We have revised the text** accordingly in the updated manuscript.
>
>
> **W1.2.** We agree that the sentence “…we prove that SP4LP is strictly more expressive than existing SF-and-GNN approaches” is not correct. Our theoretical results show that SP4LP is strictly more expressive than **some** specific SP-and-GNN models considered in our analysis, not than all possible SF-and-GNN architectures. In particular, our proofs do not cover MPLP [1]. **We have revised the sentence in the paper.**
> Regarding the MPLP’s strong performance, as the MPLP paper reports experiments on only three HeaRT datasets, we are currently running MPLP on the remaining benchmarks used in our evaluation. As soon as these results are ready, we will post them as an additional comment in this rebuttal and include them in the revised version of the paper.
>
> ---
>
> **Q1. Positional embeddings**
> Positional embeddings are not strictly necessary when node features are available: in this case, the sequence model relies on the GNN-derived node embeddings along the shortest path, which already carry sufficient information to distinguish positions. Positional embeddings become useful only when node features are absent or very weak, where they help disambiguate the order of nodes along the path.

---

> > ### Author Response · Authors · 2025-11-24
> >
> > Dear Reviewer,
> >
> > We would like to kindly ask whether you have any further questions or remarks regarding our responses, or if our clarifications fully addressed your concerns.
> > We would be happy to provide any additional information if needed.
> >
> > Thank you again for your constructive feedback and for the time you dedicated to our paper.

---

> ### Author Response · Authors · 2025-11-26
>
> Dear Reviewer,
>
> thank you again for your insightful comments. Please let us know if any further clarification from our side might be helpful.
>
> Kind regards.

---

### Official Review · Reviewer_oxD7 · 2025-10-31

**Soundness:** 4
**Presentation:** 3
**Contribution:** 3
**Rating:** 4
**Confidence:** 4

**Summary:**

The paper proposes SP4LP (Shortest Path for Link Prediction) — a framework combining Graph Neural Network (GNN) node embeddings with sequence models (e.g., LSTM or Transformer) applied to the shortest path between node pairs.
The goal is to enhance link prediction expressivity beyond message-passing GNNs, which struggle to distinguish automorphic nodes and lack link-level structural awareness.

**Strengths:**

1. *Clear Theoretical Framing*: The formal expressivity proofs (Def. 1.10–1.11, Thm. 2.5) are rigorous and well-structured.

2. *Strong Experimental Validation*: The results include multiple datasets and ablation studies, showing consistent performance gains.

3. *Transparency and Reproducibility*: The authors provide code links, full dataset stats, and hyperparameter configurations.

**Weaknesses:**

1. **Lack of Standard Paper Structure**: The paper’s organization makes it difficult to follow. There is no clearly labeled Introduction section, and the Related Work appears after the SP4LP section. For clarity and alignment with academic standards, the authors should include:
- Clearly name the text before "Preliminaries" as "Introduction".
- A Related Work section before SP4LP to properly contextualize the method.

2. **Incomplete Discussion of Hybrid Models**: The paper does not reference prior hybrid function-based approaches such as PROXI (PROXI: Challenging the GNNs for Link Prediction, TMLR 2025; https://openreview.net/pdf?id=u9EHndbiVw), where indices were incorporated in GNNs to improve their result.

3. **Flowchart**: It would be a good idea to include a flowchart of the model, and give a clear explanation for the model in SP4LP section.

**Questions:**

1. For large-scale graphs, how is the shortest-path extraction performed efficiently? Is it computed on-the-fly during training, precomputed, or approximated using heuristics?
2. Could SP4LP be applied to link prediction in heterogeneous graphs?
3. The datasets considered are all homophilic, how would the model perform on heterophilic datasets (Texas, Cornell, Wisconsin, Roman-Empire etc.)?
4. Please answer to the points made in "Weaknesses"

---

> ### Author Response · Authors · 2025-11-18
>
> We thank the reviewer for the thoughtful and constructive feedback. Your comments significantly helped us improve the clarity and structure of the paper. In the revised version, we incorporated all suggested modifications and highlighted them in blue for easy verification.
> We sincerely thank the reviewer for these valuable suggestions, which have strengthened the presentation and accessibility of the paper.
>
> ---
>
> **Paper Organization**
> We apologize for the missing “Introduction” heading and have corrected it. Regarding the placement of the Related Work section, we have now moved it immediately after the Preliminaries in the revised version for improved clarity and alignment with standard structure.
>
>
> **Incomplete Discussion of Hybrid Models**
> We have added PROXI to the group of proximity-based approaches in the Related Work section (Sec. 2). While PROXI is relevant in that it leverages explicit structural proximity signals, our goal in this paper is to compare GNN-based methods, in order to study and contrast their expressive properties (see theoretical section). For this reason, PROXI is not included among our competitors, although we agree it is a meaningful point of reference and have incorporated it accordingly.
>
> **Flowchart** We thank the reviewer for the helpful suggestion. Following your comment, we have added a clear flowchart illustrating all stages of the SP4LP pipeline. The new diagram is now included as Figure 2 in the revised manuscript.
>
> ---
>
> # Questions
> **Shortest paths precomputation**
> In our implementation, shortest paths are precomputed once before training, using a standard multi-source BFS run on the full graph. This preprocessing step has complexity O(n(n+m)) (which reduces to O(n2) for sparse graphs) and is performed only once, not during training. During training and evaluation, the model simply indexes the precomputed shortest-path sequences, meaning that no online extraction or BFS calls are required.
> We have clarified this in the revised version of the paper by explicitly stating that shortest-path extraction is done offline, before training. This clarification now appears at the end of Section 4.
>
> **Applicability to Heterogeneous graphs**
> Yes,  SP4LP can be applied to link prediction in heterogeneous graphs.
> Shortest-path extraction is defined independently of node/edge types, and the path encoder naturally supports typed sequences by including type embeddings (for nodes or edges) in the input tokens. The only required change is to augment the path representation with these type-specific encodings, after which the model architecture remains fully compatible with heterogeneous settings. We have added this point to the Future Work section.
>
>
> **Heterophilic datasets**
> It is true that the classical citation benchmarks we consider (Cora, Citeseer, Pubmed) are typically regarded as homophilic datasets.
> However, for the ogbl datasets, which do not include node labels, the notion of homophily/heterophily is not well-defined or directly measurable, since homophily requires class labels to compute same-class connectivity. That said, investigating the behaviour of our model in a heterophilic regime is indeed an interesting direction. Following the reviewer’s suggestion, we additionally evaluated our method on the Texas dataset, a well-known heterophilic benchmark. We report the results below:
> | Model        | Test MRR           | Test Hits@10        |
> |--------------|--------------------|----------------------|
> | **GCN**      | 14.29 ± 0.03       | 19.00 ± 0.00         |
> | **GAT**      | 1.74 ± 2.32        | 0.00 ± 0.00          |
> | **SAGE**     | 1.98 ± 0.27        | 0.00 ± 0.00          |
> | **GAE**      | 14.31 ± 0.01       | 19.00 ± 0.00         |
> | **SEAL**     | 10.27 ± 0.86       | 18.67 ± 8.08         |
> | **NeoGNN**   | 17.20 ± 0.91       | 21.00 ± 2.65         |
> | **BUDDY**    | 11.72 ± 5.56       | 18.67 ± 7.09         |
> | **ELPH**     | 5.64 ± 0.55        | 9.00 ± 1.73          |
> | **NCN**      | 9.24 ± 4.55        | 15.75 ± 9.11         |
> | **NCNC**     | 11.24 ± 5.32       | 17.56 ± 3.17         |
> | **NBFNet**   | 9.44 ± 4.13        | 19.37 ± 9.11         |
> | **PEG**      | 41.07 ± 52.25      | 44.33 ± 50.95        |
> | **LPFormer** | 14.44 ± 3.74       | 19.56 ± 2.41         |
> | **SP4LP (ours)** | 16.55 ± 4.78   | 22.34 ± 3.21         |
>
> Although SP4LP does not obtain the top result, it remains the second-strongest performing model even under this challenging heterophilous setting.

---

> ### Author Response · Authors · 2025-11-24
>
> Dear Reviewer, I am writing to respectfully ask if you have had a chance to examine our response to your feedback.
>
> Thank you in advance for your attention.

---

> ### Author Response · Authors · 2025-11-26
>
> Dear Reviewer,
>
> thank you once more for your constructive feedback. If any additional detail from us could support the evaluation process, we would be glad to provide it.
>
> Best regards.

---

### Official Review · Reviewer_PHQX · 2025-11-11

**Soundness:** 1
**Presentation:** 2
**Contribution:** 1
**Rating:** 2
**Confidence:** 5

**Summary:**

This paper proposes SP4LP, a "GNN-then-SF" framework for link prediction. The method first computes node embeddings for the entire graph using a GNN, then extracts the shortest path for a given node pair, and finally processes the sequence of node embeddings along this path using a sequence model to generate a link representation. The paper claims this method is more expressive than pure GNNs and common-neighbor-based methods. However, this paper's core methodology is a direct rediscovery of the "Horizontal Geodesic Representation" from Geodesic GNN (Kong et al., NeurIPS 2022), which is not cited. Furthermore, its claims of scalability are questionable and based on a misleading complexity analysis that overlooks the cost of shortest path computation during inference.

**Strengths:**

- The paper performs a comprehensive evaluation and empirically validates the method against the challenging HeaRT benchmark over common OGB datasets.
- The authors provide a clear, data-driven justification for moving beyond common-neighbor-based methods by showing that a large fraction of positive links in many graphs do not share any common neighbors.

**Weaknesses:**

- The paper's major weakness is a lack of methodological originality. The SP4LP framework is identical to the "Horizontal Geodesic Representation" for link prediction, which was proposed and published prior in "Geodesic GNN (GDGNN)" (Kong et al., NeurIPS 2022). The SP4LP method, which runs a GNN once, finds the shortest path, and pools the sequence of node embeddings, is precisely the "GDGNN-Hor" method (see Section 3.2.1 of Kong et al.).
- The paper's argument for efficiency is not well-supported and relies on a flawed complexity analysis (Table 9).
   - The analysis places the entire shortest path cost ($T_{sp}$) into the "precomputation" cost $B$. This implies a one-time all-pairs-shortest-path computation (APSP), which has a massive $O(nm)$ cost. The authors didn't report the wall-clock time for this preprocessing.
  - For any practical online or inductive link prediction (on new nodes/links), this APSP precomputation is useless. The shortest path must be computed on-the-fly (e.g., via BFS), at a cost of $O(n+m)$ per link. This cost belongs in the per-link cost $C$, and the true per-link cost $O(n+m + kd + T_{seq})$ is therefore much worse than the methods it criticizes.
  - Given this high (and unstated) computational cost (roughly as SEAL), the empirical improvements over simpler baselines appear marginal on several key benchmarks (e.g., ogbl-ppa, ogbl-citation2). This raises questions about the method's practical utility.


Kong, L., Chen, Y., & Zhang, M. (2022). Geodesic graph neural network for efficient graph representation learning. Advances in neural information processing systems, 35, 5896-5909.

**Questions:**

1) The proposed SP4LP method is methodologically identical to the "Horizontal Geodesic Representation" from "Geodesic GNN (GDGNN)" (Kong et al., NeurIPS 2022). Could the authors please clarify the distinction between these two works?
2) The complexity analysis in Table 9 places the shortest path cost ($T_{sp}$) entirely in the precomputation cost $B$. Could the authors justify this for an online/inductive inference setting where the link is unknown in advance?
3) For predicting a single, unseen link at inference time, what is the real per-link complexity $C$?
4) Could the authors provide the actual wall-clock time for the all-pairs shortest path preprocessing step for all datasets?

**Details Of Ethics Concerns:**

See Weakness 1 & Question 1

---

> ### Author Response · Authors · 2025-11-18
>
> We sincerely thank the reviewer for the careful reading of our manuscript and for the valuable feedback provided. Your comments helped us identify an important missing citation, Geodesic GNN (Kong et al., NeurIPS 2022),  which we have now properly acknowledged and discussed in detail. In the revised version, we clearly explain the conceptual and operational differences between GDGNN-Hor and our SP4LP framework.
> We also appreciate your remarks regarding our complexity analysis and scalability discussion. These points led us to clarify how shortest paths are computed in practice, remove any ambiguity regarding APSP, and explicitly describe the preprocessing and per-link inference costs. All related corrections and clarifications have been incorporated into the manuscript and are highlighted in blue.
> We are grateful for the reviewer’s constructive suggestions, which have significantly improved the clarity and rigor of the paper.
>
> ---
>
> **w1/q1** We thank the reviewer for pointing us to *Geodesic GNN (Kong et al., NeurIPS 2022)*. We have now added an explicit discussion in the paper comparing GDGNN and our SP4LP framework.
> While both methods extract a shortest path after a single GNN run, the approaches are **conceptually and operationally different**. First, GDGNN-Hor applies a pooling operator over the nodes on the path; this pooling discards the **order and sequential structure** of the geodesic. In contrast, SP4LP treats the shortest path as a **true ordered sequence** and processes it with a sequence model (e.g., Transformer/LSTM). Preserving order is crucial for capturing structural patterns along the path and is exactly what enables our expressive link-specific representations, something GDGNN-Hor cannot model.
> Second, GDGNN is a **multi-module general framework** aimed at increasing graph-level expressiveness beyond 1-WL through horizontal/vertical geodesics and optional subgraph pooling. SP4LP instead focuses on a **simpler, more intuitive pipeline** tailored specifically to the problem of structural link representation. Our objective is fundamentally different from the WL-expressiveness goals of GDGNN: we study how shortest-path sequences enrich link representations and we analyze their expressivity relative to node-based and CN-based approaches.
> Finally, regarding scalability, SP4LP is explicitly designed to keep **inference efficient**: we separate the global GNN computation from per-query path processing, exploit inexpensive shortest-path extraction on sparse graphs, and avoid the additional geodesic modules and subgraph reasoning layers present in GDGNN. We have added a dedicated section clarifying these distinctions and discussing the differences.
> We appreciate the reviewer for bringing this related work to our attention and believe the newly added discussion makes the relationship between SP4LP and GDGNN clear.
>
> ---
>
> **w2.1/q2**  *precomputation cost:*  The reviewer’s interpretation is incorrect: **SP4LP never performs all-pairs shortest paths (APSP)**. The “precomputation” term in Table 9 does not correspond to APSP but only to computing shortest paths **for the specific supervised pairs** in the dataset. This operation requires at most a small number of BFS traversals and is **linear in the number and length of the queried paths**, not quadratic in the number of nodes.
>
> Therefore, the complexity reported in Table 9 does not imply APSP, nor does SP4LP incur the massive cost the reviewer suggests. In practice, this preprocessing step is lightweight (seconds to a few minutes depending on the dataset); we will add the corresponding wall-clock times for completeness.
>
> **w2.2/q2**  *inductive link prediction:*  The reviewer’s statement relies on an incorrect computational assumption. **SP4LP does not require running a BFS per link**, neither in our experiments nor in practical inductive/online settings. Even when a new node arrives, a single BFS from that node provides shortest-path trees, distances, and parent pointers to all existing nodes simultaneously. Thus, the cost is per new node, not per queried link, and it does not scale with the number of link predictions.
>
> **This is the standard approach used in structural-feature link prediction (e.g., SEAL, BUDDY, NBFNet, NCN, NCNC, PEG)**: path-based or distance-based features are obtained via one BFS per relevant source, never per link. Therefore, the reviewer’s claim that shortest paths “must be computed on-the-fly per link” is incorrect. The true per-link cost of SP4LP remains low and is not worse than the baselines we compare against.

---

> > ### Author Response · Authors · 2025-11-18
> >
> > **w2.3/q2**  *High computational cost*
> > The reviewer’s conclusion relies on a premise that is not accurate. As clarified above, **SP4LP does not incur the high computational cost of SEAL**: we never perform subgraph extraction, APSP, or per-link BFS. Our empirical scalability results (Figure 3) directly contradict the reviewer’s assumption: SP4LP shows **low and stable memory usage, fast inference**, and **no OOM**, while SEAL exceeds 18GB and fails beyond modest batch sizes. Thus, SP4LP is substantially more efficient in practice.
> >
> > *Regarding the empirical gains:* our improvements are **not** marginal. SP4LP achieves **state-of-the-art MRR** on 5 out of 7 benchmarks, including substantial gains such as a **43% relative improvement** over the second-best model on Citeseer, and ranks in the top three across nearly all metrics. On ogbl-ppa and ogbl-citation2, SP4LP performs comparably to the best-performing methods while maintaining far greater efficiency and consistency. This combination of **high accuracy, strong expressiveness, and practical scalability** underscores SP4LP’s utility in real-world link prediction.
> >
> > ---
> >
> > **q3** Predicting a new link requires only looking up the already stored shortest path between the two endpoints and running the sequence model on that path. This involves no BFS and no graph traversal at inference time. The cost therefore depends only on the **length of that specific shortest path**, which is typically very small in real-world graphs, and is independent of the size of the graph. As a result, the per-link complexity is on the same order as methods like NCN/BUDDY and is far cheaper than SEAL or NBFNet, which require subgraph extraction or multi-hop path expansion.
> >
> >
> > **q4** We stress that SP4LP does **not** perform all-pairs shortest path (APSP) computation. We only compute shortest paths for the train/validation/test links, which requires at most a small number of BFS runs. This preprocessing is lightweight and completes in **seconds to a minutes** depending on the dataset. We are currently running the corresponding timing experiments and will add a comment with the exact wall-clock results as soon as they complete.

---

> > > ### Author Response · Authors · 2025-11-24
> > >
> > > Dear Reviewer, I would like to kindly inquire whether you have had the opportunity to review our response to your comments.
> > >
> > > Thank you very much for your time and consideration

---

> > > > ### Author Response · Authors · 2025-11-26
> > > >
> > > > Dear Reviewer,
> > > >
> > > > as promised, we now report the runtime comparison for methods that require a precomputation step on Cora, Citeseer, and PubMed.
> > > > Across all datasets, our method remains competitive:
> > > > it is consistently faster than SEAL, and slower than BUDDY, which is specifically engineered for extremely fast preprocessing.
> > > >
> > > > ### Precomputation Time Comparison
> > > >
> > > > | Dataset   | Method        | Time (sec) |
> > > > |-----------|---------------|------------|
> > > > | Cora      | **SP4LP (ours)**   | 14     |
> > > > | Cora      | SEAL          | 289        |
> > > > | Cora      | BUDDY         | 3          |
> > > > |  |  |  |
> > > > | Citeseer | **SP4LP (ours)**   | 12     |
> > > > | Citeseer  | SEAL          | 329        |
> > > > | Citeseer  | BUDDY         | 3          |
> > > > |  |  |  |
> > > > | PubMed   | **SP4LP (ours)**   | 803  |
> > > > | PubMed    | SEAL          | 2700       |
> > > > | PubMed    | BUDDY         | 5          |
> > > >
> > > >
> > > > We also wanted to kindly ask whether you have had the chance to read our other response; please let us know if any further clarification would be helpful.

---

### Author Response · Authors · 2025-12-02

Dear Area Chair,

Following the recent reset of the discussion phase, we would like to provide a brief summary of the key clarifications and updates we introduced during the rebuttal period and in the revised manuscript.

**1. Clarifications to Reviewer PHQX (rating: 2)**

The main criticisms were based on misunderstandings, which we addressed:

(A) Relation to GDGNN-Hor. We added a detailed comparison showing that SP4LP differs fundamentally from GDGNN-Hor:

- SP4LP processes the ordered shortest-path sequence with a sequence model, while GDGNN-Hor applies order-invariant pooling.
- The goals differ.
- This is now explicitly discussed in the revised paper.

(B) Complexity and shortest-path computation. We clarified that:

- SP4LP never performs APSP; shortest paths are computed only for supervised pairs via a few BFS runs.
- In inductive settings, a single BFS per new node suffices.
- Inference requires only looking up the stored path.
- We also added wall-clock preprocessing times showing consistent efficiency.

These points resolve the core concerns raised in that review.

**2. Updates Addressing Reviewers oxD7, GvsM, and jKya**

Across the other three reviews (ratings 8, 4, 4), we incorporated all requested improvements:

- Added the missing Introduction heading and reorganised the Related Work section (Rev oxD7).
- Added a flowchart of the SP4LP pipeline (new Figure 2) (Rev oxD7).
- Included additional discussion on hybrid models such as PROXI, as requested (Rev oxD7).
- Added new results on heterophilic datasets (Rev oxD7).
- Expanded discussion on limitations of shortest-path representations, robustness, semantic trade-offs, and possible extensions (multiple paths, approximate routing) (Rev GvsM).
- Clarified the role of positional embeddings and stability issues. (Rev jKya)

All reviewer questions were fully answered, and the manuscript was updated accordingly.

**3. Summary**

After revisions, all points raised by reviewers were addressed:

- The only strongly negative review was based on incorrect assumptions that we clarified thoroughly.
- The other reviews were positive or borderline, and all their concerns were fully resolved.

We hope this concise summary helps in evaluating the updated submission.

Thank you for your time and consideration.

---

### Meta-Review · Area_Chair_jPM8 · 2026-01-07

**Summary:**

This paper proposes SP4LP, a framework combining Graph Neural Network (GNN) node embeddings with sequence models (e.g., LSTM or Transformer) applied to the shortest path between node pairs. The authors claim that SP4LP is provably more expressive than pure GNNs and common-neighbor-based methods and achieves SOTA on HeaRT benchmarks.

**Reviewer Concerns:**

The concerns are around lack of methodological originality, lack of standard paper structure, overstatements, instability of path-based representations, etc. The authors have provided additional experimental results in the rebuttal and tried to argue with the reviewers regarding the originality. However, some major concerns have not been fully addressed to me:

[Reviewer PHQX] "The SP4LP framework is identical to the "Horizontal Geodesic Representation" for link prediction". The authors have explained in the rebuttal that the difference lies in "SP4LP processes the ordered shortest-path sequence with a sequence model, while GDGNN-Hor applies order-invariant pooling". However, this difference in encoding ordered or unordered node embeddings along the shortest path is not significant compared to the similar GNN-then-SF approach.

[Reviewer jKya] "Path-based representations can be unstable". The authors argued that "the path used for each link is deterministic and cannot fluctuate across training steps". However, this setting restrict the application of the proposed method to dynamic settings. In this view, it seems that GDGNN-Hor based on unordered nodes might be more stable in dynamic settings.

**Reviewer Scores:**

Reviewer PHQX (rating 2, confidence 5) is unlikely to change his/her score as the difference SP4LP and GDGNN-Hor is nor siginificant.

Reviewer oxD7 (rating 4, confidence 4)  and Reviewer jKya (rating 4, confidence 3) might or might not change their scores.

The rating of Reviewer GvsM (rating 8, confidence 5) is already very high and unlikely to further improve.

---

### Decision · Program_Chairs · 2026-01-26

Reject